# Role-Level Inductive Bias for Cross-Task Generalization in Multi-Agent Reinforcement Learning

**Chang Yao** [1 2]  **Youfang Lin** [1 2]  **Shoucheng Song** [1 2]  **Hao Wu** [1 2]  **Shengkun Yang** [1 2]  **Yuqing Ma** [3]  **Kai Lv** [1 2]

## Abstract

Achieving cross-task generalization remains a critical challenge in Multi-Agent Reinforcement Learning (MARL), fundamentally relying on effective inductive biases. However, existing entity-level biases often overlook collaborative patterns, whereas task-level biases lack sufficient coverage for novel scenarios. To address this, we introduce a role-level inductive bias as an intermediate abstraction that integrates entity-level flexibility with task-level inter-agent collaboration. To instantiate this, we propose Gaussian-mixture-model-based Transferable Role discovery (GTR). Specifically, GTR constructs a structured role space to ensure diverse role assignment, further achieves role decoupling via regularization, and ultimately utilizes these roles for efficient generalization. Empirical results demonstrate that GTR achieves superior zero-shot and few-shot transfer performance on unseen tasks compared to state-of-the-art methods.

## 1. Introduction

Multi-agent Reinforcement Learning (MARL) shows significant promise in domains like robot swarms (Lv et al., 2025) and traffic control (Ruan et al., 2024). However, current methods mostly focus on fixed tasks and struggle to generalize to unseen scenarios, particularly those involving dynamic team size (Rashid et al., 2020; Lowe et al., 2017; Wu et al., 2025a; Song et al., 2025). This lack of generalization ability (Yao et al., 2025) constrains agents' adaptability to dynamic task requirements in the real world, thus severely hindering the practical deployment of multi-agent systems.

[1]School of Computer Science & Technology, Beijing Jiaotong University, Beijing, China [2]Beijing Key Laboratory of Traffic Data Mining and Embodied Intelligence, Beijing, China [3]Institute of Artificial Intelligence, Beihang University, Beijing, China. Correspondence to: Kai Lv <lvkai@bjtu.edu.cn>.

*Proceedings of the 43rd International Conference on Machine Learning*, Seoul, South Korea. PMLR 306, 2026. Copyright 2026 by the author(s).

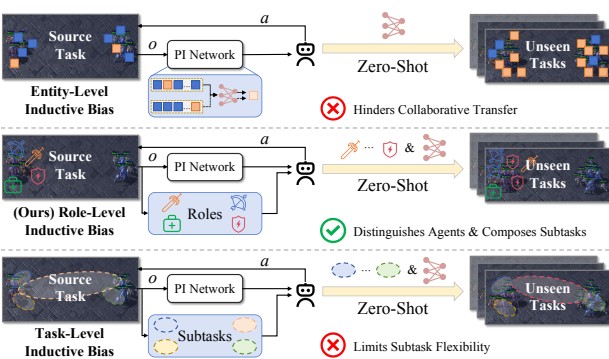

*Figure 1.* An overview of inductive bias at three levels to achieve multi-agent generalization. PI denotes permutation invariance.

Existing methods mainly employ inductive biases to guide agents in learning transferable policies, enabling generalization to unseen tasks. As shown in Figure 1, such works generally operate at two levels. At the **entity-level** (low-level), methods typically employ Permutation Invariance (PI) to generalize individual policies (Hu et al., 2021b; Jianye et al., 2022). Despite offering flexibility with entity scale, PI often fails to preserve inter-agent collaborative patterns for cross-task generalization. In contrast, at the **task-level** (high-level), methods explicitly decompose tasks into reusable sub-task modules (Tian et al., 2023; Wu et al., 2025b). Although sub-tasks consider inter-agent coordination, their limited coverage typically leads to failure in complex scenarios with novel sub-tasks. To this end, we propose a **role-level** inductive bias serving as an **intermediate abstraction** between entity- and task-level biases.

This role-level bias effectively integrates entity-level flexibility with task-level transferable collaboration patterns to enhance cross-task generalization. On one hand, roles serve as functional distinctions, endowing agents with specific responsibilities within the team to facilitate coordination. On the other hand, the flexible composition of roles overcomes the limited coverage inherent in rigid sub-task modules. Under this role-level inductive bias, roles exhibit dual efficacy: at the task-level, enabling flexible collaboration patterns via global role reassignment; at the entity-level, empowering individual agents to rapidly anchor their functional positioning and efficiently reuse prior policies.

Guided by this role-level bias, we revisit existing role-based methods. While certain approaches infer roles via clustering or contrastive learning on trajectories and actions (Wang et al., 2020a;b; Hu et al., 2023; Goel et al., 2025) to encourage behavioral diversity within fixed scenarios, they falter when generalizing to novel tasks. Specifically, rigid clustering leads to a misalignment between role assignment and task demands, failing to accommodate dynamic team compositions. Furthermore, the excessive coupling between role embeddings and specific environmental details significantly limits generalization performance in unseen scenarios.

In this paper, we propose **G**aussian mixture model-based **T**ransferable **R**ole discovery (GTR), a framework that realizes role-level inductive bias. Treating roles as transferable collaborative knowledge, GTR boosts cross-task generalization through two core mechanisms designed for team adaptation and role reusability. First, to ensure robust role assignment against shifting team compositions, we employ a Permutation-Invariant Role Encoder integrated with a Utility-Weighted Determinantal Point Process (UW-DPP) (Kulesza et al., 2012). This combination dynamically selects functionally diverse roles from the GMM-based latent space, preventing role collapse while maintaining coverage. Second, to guarantee cross-task reusability, we introduce a dual-stream regularization that predicts intentions and influences. This design decouples role embeddings from specific environmental details, enabling agents to apply learned capabilities to novel scenarios. Finally, we incorporate roles into decision-making and credit assignment to facilitate GTR's cross-task generalization. Our main contributions are:

- We introduce a role-level inductive bias as an intermediate abstraction between entity- and task-level biases, facilitating policy transfer across diverse scenarios.

- We propose GTR, which leverages a GMM prior to construct a latent role space. By integrating diverse role assignment and efficient role regularization, GTR overcomes the limitations of existing methods.

- Empirical results show that GTR excels in both zero-shot generalization and few-shot transfer tasks, significantly outperforming baselines on SMAC (Samvelyan et al., 2019) and SMACv2 (Ellis et al., 2023).

## 2. Related Work

### 2.1. Generalization in MARL

**Entity-Level Inductive Bias via Permutation Invariance.** This category of methods enforces Permutation Invariance (PI) within policy architectures to accommodate varying team scales. UPDeT (Hu et al., 2021b) and HPN (Jianye et al., 2022) enforce PI via Transformers and Hypernetworks, decoupling policies from input dimensions. E2GN2

(McClellan et al., 2024) combines geometric priors to achieve zero-shot generalization using an Equivariant GNN. In addition, ASN (Yang et al., 2023) and RPG (Yao et al., 2025) focus on mapping entity-level observations to task-invariant patterns. However, PI overlooks collaborative knowledge transfer among agents, thereby limiting its adaptability to complex changes in team composition.

**Task-Level Inductive Bias via task decomposition.** These methods assume diverse tasks share decomposable structures that can be explicitly transferred. One extracts reusable skills: BiKT (Zhang et al., 2026) and HyGen (Zhang et al., 2024) learn to combine skills or tactic codebooks from offline data, while (Korte et al., 2025) encapsulates capabilities into reusable causal macros. Another focuses on task decomposition (Tian et al., 2023; Zeng et al., 2024), guiding adaptation by discovering generalizable sub-tasks or sub-goals. Additionally, MATTAR (Qin et al., 2024) learns task embeddings to transfer knowledge based on dynamic similarity. While effective for coordinating knowledge reuse, these methods often fail to compose diverse collaborative patterns, leading to failure in scenarios with novel sub-tasks.

Beyond team scales and compositions, recent studies explore MARL generalization across teammate policies and environments. The former emphasize ad-hoc teamwork and zero-shot coordination, adapting to novel partners (Li et al., 2026). Concurrently, the latter enhance robustness by generating diverse scenarios (Jha et al., 2025).

### 2.2. Role Discovery in MARL

**Observation and Trajectory-based Role Discovery.** These approaches learn latent role representations from observations or trajectories. ROMA (Wang et al., 2020a) maps observations to a stochastic role space, enabling specialized policies via implicit clustering. This concept is extended by ROMAT (Wang et al., 2024) to offline settings, utilizing behavior cloning to mitigate collaborative conflicts. To capture temporal dynamics, ACORM (Hu et al., 2023) applies contrastive learning to agent trajectories. Diverging from historical dependence, R3DM (Goel et al., 2025) defines roles by future behaviors, leveraging world models to generate intrinsic rewards based on predictive trajectories.

**Action-based Role Discovery.** Alternatively, roles can be defined by action clustering. RODE (Wang et al., 2020b) clusters agents based on the environmental impact of their actions, restricting action spaces to enforce distinct roles. Similarly, SIRD (Zeng et al., 2023) and DRDA (Xia et al., 2023) utilize hierarchical clustering or action contribution analysis to structure role definitions implicitly.

However, role-based methods fail to handle variable-length inputs and tend to overfit role representations to specific tasks, thereby hindering knowledge transfer.

To address these limitations, we propose a role-level inductive bias to bridge the gap between entity-level and task-level biases. Our framework, GTR, achieves cross-task generalization via transferable role discovery.

# 3. Preliminaries

Scalable multi-agent coordination tasks are formulated as a transferable role-based Decentralized Partially Observable Markov Decision Process (Dec-POMDP) $\mathcal{G} = \langle \mathcal{I}, \mathcal{S}, \mathcal{A}, P, R, O, \Omega, \gamma \rangle$, where $\mathcal{I} = \{1, \ldots, N\}$ is the set of $N$ agents, $\mathcal{S}$ is the global state space, and $O = \times_{i \in \mathcal{I}} O^i$ denotes the joint observation space. The joint action space is $\mathcal{A} = \times_{i \in \mathcal{I}} \mathcal{A}^i$, where $\mathbf{a} = (a^1, \ldots, a^N)$ represents the joint action. We distinguish between source tasks $\mathcal{T}_{src}$ for training and target tasks $\mathcal{T}_{tgt}$ for evaluation. The number of agents $N$ varies between domains ($N_{src} \neq N_{tgt}$), scaling the dimensions of $\mathcal{S}$, $O$, and $\mathcal{A}$. At timestep $t$, the state transitions via $P(s_{t+1} \mid s_t, \mathbf{a}_t)$ with global reward $r_t = R(s_t, \mathbf{a}_t)$. Agent $i$ receives a local observation $o_t^i \in O^i$ determined by the function $\Omega(s_{t+1}, i)$. Here, the discount factor is $\gamma \in [0, 1]$. In addition, each agent instantiates a discrete role $c_t^i \in \{1, \ldots, K\}$ and a corresponding role representation $z_t^i$. Here, $K$ is the number of roles. The agent then operates under a policy conditioned on $\tau_t^i = (o_t^i, a_{t-1}^i)$, $c_t^i$, and $z_t^i$:

$$a_t^i \sim \pi_\theta(\cdot \mid \tau_t^i, c_t^i, z_t^i). \quad (1)$$

We adopt the Centralized Training with Decentralized Execution (CTDE) paradigm (Kraemer & Banerjee, 2016). During training, the algorithm utilizes the global state $s_t$ to guide the learning, while execution relies solely on local histories. Our objective is to optimize parameters $\theta$ to maximize the expected return on the source tasks $\mathcal{T}_{src}$, formulated as $J(\theta) = \mathbb{E}_{\mathcal{G} \sim \mathcal{T}_{src}}[\mathbb{E}_{\pi_\theta}[\sum_{t=0}^{\infty} \gamma^t r_t]]$.

# 4. Methodology

In this section, we present the proposed GTR framework, which incorporates a role-level inductive bias to achieve multi-agent cross-task generalization. As illustrated in Figure 2, GTR operates through three key phases: (1) **Role Assignment** (Section 4.1): We establish a universal role space that remains valid across varying team scales. This phase assigns functionally diverse roles to agents to prevent strategy collapse in complex scenarios. (2) **Role Regularization** (Section 4.2): Role representations should be decoupled from task-specific details to serve as transferable primitives. We purify these roles by focusing on capturing agent intentions and influences, ensuring efficient transfer. (3) **Role Utilization** (Section 4.3): Finally, we inject the roles into both policy and credit assignment. This fosters behavioral diversity and ensures adaptability to unseen tasks.

## 4.1. Role Assignment

To achieve scalable coordination, we formalize role discovery as a probabilistic inference problem. Specifically, GTR first encodes the agent's local trajectory into a permutation-invariant representation and then projects it onto a structured latent manifold governed by a Gaussian Mixture Model (GMM) representing distinct behavioral modes.

### 4.1.1. PERMUTATION-INVARIANT ENCODING

To handle dynamic team scales and unstructured observations, we require an encoding scheme that is insensitive to the ordering and quantity of observed entities. Unlike fixed-size vector inputs, the local observation $o_t^i$ is decomposed into the agent's own attributes, $o_{t,i}^{ego}$, and the set of attributes corresponding to other observable entities, $o_{t,i}^{others}$. To capture the permutation-invariant embedding $x_t^i$ without overfitting to specific entity indices, we employ an attention mechanism as a set function approximator.

Specifically, we fuse the current ego-features with the previous action to construct the query $\mathbf{q}_i = f_q([o_{t,i}^{ego}, a_{t-1}^i])$. Concurrently, the features of other entities are projected to form the keys $\mathbf{K} = f_k(o_{t,i}^{others})$ and values $\mathbf{V} = f_v(o_{t,i}^{others})$. The final permutation-invariant embedding $x_t^i$ is derived via a residual cross-attention block:

$$x_t^i = \text{LayerNorm}\left(\mathbf{q}_i + \text{softmax}\left(\frac{\mathbf{q}_i \mathbf{K}^\top}{\sqrt{d_k}}\right)\mathbf{V}\right). \quad (2)$$

This formulation ensures that $x_t^i$ remains consistent regardless of the number or permutation of entities, enabling zero-shot transfer across varying team scales.

### 4.1.2. STRUCTURED VARIATIONAL ROLE INFERENCE

Based on the embedding $x_t^i$, we seek to infer a role that explicitly disentangles discrete functional modes from their continuous intra-mode variations. We capture this compositional structure by modeling the role space with a GMM. Formally, a role comprises a discrete variable $c_t \in \{1, \ldots, K\}$ and a continuous variable $z_t \in \mathbb{R}^d$.

**Prior: Latent Space Structure.** To structure the latent space, we define a GMM prior consisting of: (1) a uniform categorical distribution $p(c_t)$ to encourage role diversity, and (2) conditional Gaussian distributions $p(z_t|c_t)$ centered at learnable $\boldsymbol{\mu}_k$ to capture intra-role variability.

$$p(c_t) = \text{Cat}(1/K), \quad p(z_t|c_t = k) = \mathcal{N}(\boldsymbol{\mu}_k, \boldsymbol{\Sigma}_k). \quad (3)$$

Here, $\{\boldsymbol{\mu}_k\}_{k=1}^K$ ensure that trajectories with similar functional properties are clustered together.

**Inference: Stochastic Embedding.** Formally, we approximate the posterior using a factorized inference network

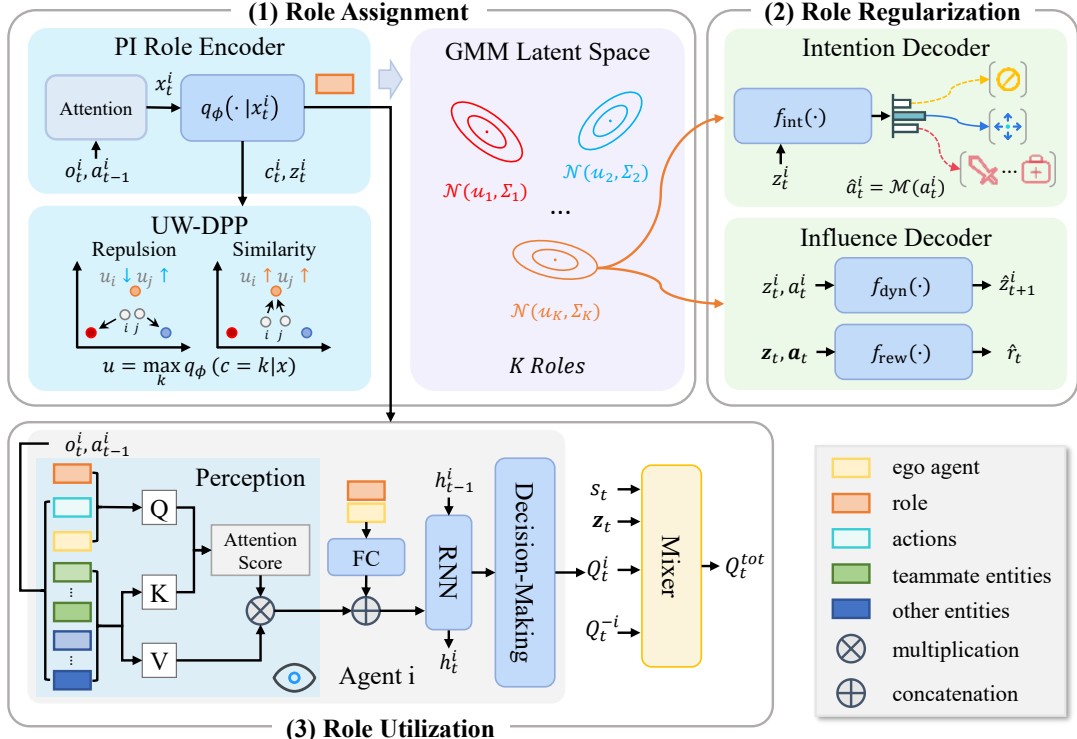

*Figure 2.* Overview of the GTR architecture. The framework consists of three main parts: (1) *Role Assignment*, which includes the PI Role Encoder, GMM Latent Space, and UW-DPP; (2) *Role Regularization*, comprising the Intention Decoder and Influence Decoder; and (3) *Role Utilization*, involving the Agent and Mixer networks.

$q_\phi(z_t, c_t | x_t) = q_\phi(c_t | x_t) q_\phi(z_t | x_t, c_t)$. To enable backprop-agation through the continuous latent variable, we use the reparameterization trick. The inference network outputs the distribution parameters $(\boldsymbol{\mu}_\phi, \boldsymbol{\sigma}_\phi)$, and $z_t$ is sampled as:

$$z_t = \boldsymbol{\mu}_\phi(x_t, c_t) + \boldsymbol{\sigma}_\phi(x_t, c_t) \odot \boldsymbol{\epsilon}, \quad \boldsymbol{\epsilon} \sim \mathcal{N}(0, \mathbf{I}), \quad (4)$$

which makes the sampling process differentiable, allowing the model to learn the distribution parameters end-to-end.

**Structural Constraint.** To ensure a well-defined structure within the latent space, we impose a regularization loss based on the KL divergence between the posterior and the GMM prior. This constrains the learned representations to align with the predefined latent structure:

$$\mathcal{L}_{\text{KL}} = \underbrace{D_{\text{KL}}(q_\phi(c_t|x_t) \| p(c_t))}_{\text{Cluster Balancing}}$$
$$+ \sum_{k=1}^{K} q_\phi(c_t = k | x_t) \underbrace{D_{\text{KL}}(q_\phi(z_t|x_t, k) \| p(z_t|k))}_{\text{Intra-Cluster Alignment}}.$$
$$(5)$$

Here, the *Cluster Balancing* term penalizes deviations from the uniform prior to prevent mode collapse, while the *Intra-Cluster Alignment* term pulls embeddings towards their assigned prototypes $\boldsymbol{\mu}_k$, ensuring that the learned representations are compact and distinct.

### 4.1.3. DIVERSE ROLE ASSIGNMENT VIA UW-DPP

Diverse role assignment is crucial for robust generalization, enabling the team to adapt to unseen scenarios by recomposing learned roles. While standard Determinantal Point Processes (DPPs) (Kulesza et al., 2012) effective enhance diversity in domains like recommender systems by penalizing feature similarity (see Appendix B). However, directly applying DPPs leads to "Blind Repulsion": indiscriminately forcing orthogonal representations hinders cooperative behaviors requiring high consistency (e.g., focused fire).

To address this, we propose Utility-Weighted DPP (UW-DPP) to balance diversity with cooperative utility. We define the role confidence $u_i = \max_k q_\phi(c_t^i = k | x_t^i) \in [0, 1]$ as a measure of the agent's utility in its current role. We construct the dynamic kernel matrix $\mathbf{L}_t \in \mathbb{R}^{N_t \times N_t}$ with entries combining utility and similarity:

$$L_{ij} = b \cdot (u_i u_j)^2 \cdot \mathcal{K}(z_t^i, z_t^j), \quad (6)$$

where $\mathcal{K}(\cdot, \cdot)$ is the kernel function (e.g., RBF kernel) measuring the similarity, and $b$ is a scaling factor.

Geometrically, maximizing $\det(\mathbf{L}_t)$ corresponds to maximizing the volume of the parallelotope spanned by the feature vectors (Kulesza & Taskar, 2011). This induces a self-regulating trade-off: in early training, uncertain agents are

forced toward orthogonality to maintain volume, effectively driving exploration by penalizing similarity. As the policy converges, role assignments stabilize and the confidence $u_i$ increases accordingly. This expanded volume budget accommodates higher feature similarity, thereby facilitating the role redundancy required for complex coordination.

Notably, this volumetric constraint is size-invariant, naturally adapting to dynamic agent numbers $N_t$. To enforce this constraint during training, we minimize the negative log-determinant loss:

$$\mathcal{L}_{\text{DPP}} = -\frac{1}{N_t} \log \det(\mathbf{L}_t + \xi \mathbf{I}), \qquad (7)$$

where $\mathbf{I}$ is the identity matrix, and $\xi$ is a small scalar added for numerical stability.

## 4.2. Role Regularization

While the constraints in Section 4.1 ensure a well-structured latent space, they do not guarantee that the representations capture information essential for cross-task generalization. To ensure the functional utility of the inferred roles, we regularize the latent space through two complementary decoding tasks: Intention Prediction (what the agent wants to do) and Influence Prediction (what effect the agent causes).

### 4.2.1. INTENTION: ACTION ABSTRACTION

Standard reconstruction objectives often compel the role to memorize task-specific actions, which limits generalization. To extract transferable intentions, we regularize the role space by predicting coarse-grained functional categories.

We define a surjection $\mathcal{M} : \mathcal{A} \rightarrow \mathcal{A}_{abs}$ that maps atomic actions to universal abstract categories. Specifically, we leverage the inherent structure common across MARL domains to group actions into three logical modes: (i) *Idle* (Passivity), (ii) *Self-Directed* (Own attribute updates), and (iii) *Interaction* (Directed at other entities). The Intention Decoder $f_{\text{int}}$ estimates the probability of these categories given the role embedding $z_t$. The objective is to maximize the log-likelihood of the abstract label $\hat{a}_t = \mathcal{M}(a_t)$:

$$\mathcal{L}_{\text{int}} = \mathbb{E}_{q_\phi} \left[ -\log p(\hat{a}_t | z_t) \right]. \qquad (8)$$

This objective compels $z_t$ to discard task-specific execution variations while retaining the core intent, facilitating zero-shot transfer even when the atomic action space $\mathcal{A}$ changes.

### 4.2.2. INFLUENCE: ROLE DYNAMICS & EFFICACY

Beyond describing immediate intentions, a valid role representation must capture the agent's influence on the future. We implement this principle by modeling both local role evolution and global team efficacy.

**Role Dynamics Consistency.** To ensure the role representation is predictive of its future evolution conditioned on the current action, we introduce a role transition predictor $f_{\text{dyn}}$ that estimates the future role distribution conditioned on the current role sample $z_t$ and action $a_t$:

$$\hat{p}(z_{t+1} | z_t, a_t) = \mathcal{N}(\hat{\boldsymbol{\mu}}_{t+1}, \hat{\boldsymbol{\sigma}}_{t+1}^2), \qquad (9)$$

where $(\hat{\boldsymbol{\mu}}_{t+1}, \hat{\boldsymbol{\sigma}}_{t+1}^2) = f_{\text{dyn}}(z_t, a_t)$. To ensure the role embedding captures the effect of $a_t$, we align this predicted distribution with the posterior encoded from the next timestep. This forms a self-supervised consistency objective:

$$\mathcal{L}_{\text{dyn}} = \mathbb{E}_{q_\phi} \left[ D_{\text{KL}} \big( \text{sg}[q_\phi(z_{t+1} | x_{t+1})] \parallel \hat{p}(z_{t+1} | z_t, a_t) \big) \right], \qquad (10)$$

where $\text{sg}[\cdot]$ denotes the stop-gradient operator. By stopping gradients on the target, we prevent representation collapse, forcing the predictor to learn role dynamics.

**Global Efficacy.** To ground individual roles in the team's collective goal, we predict the global reward $r_t$. Recognizing that rewards emerge from joint interactions, we aggregate individual roles into a mean-pooled team context $\bar{z}_t = \frac{1}{N} \sum_i z_t^i$. The reward prediction objective is defined as the mean squared error:

$$\mathcal{L}_{\text{rew}} = \mathbb{E}_{q_\phi} \left[ \| r_t - f_{\text{rew}}(\bar{z}_t, \mathbf{a}_t) \|^2 \right]. \qquad (11)$$

We combine the dynamics and reward objectives into a unified influence term: $\mathcal{L}_{\text{inf}} = \mathcal{L}_{\text{rew}} + \mathcal{L}_{\text{dyn}}$. Jointly optimizing these terms ensures that $z_t$ encodes both local dynamics and the agent's contribution to the global outcome.

## 4.3. Role Utilization

With the discrete role $c_t^i$ and continuous role embeddings $z_t^i$, we incorporate them into the RL loop to guide both policy learning and value function estimation.

**Role-Conditioned Perception and Decision-Making.** To utilize the discovered roles, we integrate the joint role representation $[c_t^i, z_t^i]$ into the agent's policy through two distinct mechanisms: acting as a perceptual filter and ensuring temporal consistency. First, within the attention-based perception module (as shown in Figure 2), the role representation is integrated into the query generation. This formulation serves as a cognitive filter, directing the agent's attention toward role-relevant entities—for instance, causing a "Healer" to prioritize "injured allies". Second, to maintain temporal consistency, the role representation is fed into the RNN alongside the processed observations. This ensures the agent's policy remains aligned with its intended role. Finally, these role-conditioned features are passed to the decision-making head to generate the action values $Q_t^i$.

**Role-Based Credit Assignment.** Standard mixing networks rely on the global state $s_t$ to estimate the joint action-value

$Q_{tot}$. However, the dimension of $s_t$ varies with the number of entities, preventing transfer. Instead of raw states or mean-pooling, we construct a permutation-invariant global embedding by aggregating agent-specific features reinforced by roles. Specifically, for each agent, we concatenate its attributes with its role embedding $z_t^i$, and then employ an attention mechanism to aggregate these distinct functional contributions into a global context vector (similarly applied to enemy features). This aggregation ensures that the credit assignment captures the complex interplay between diverse roles regardless of team scale. The detailed architecture of this mixing module is provided in Appendix C.

### 4.4. Joint Optimization Objective

The GTR framework is trained end-to-end by minimizing the following cumulative loss function, which balances the reinforcement learning goal with structured representation learning. The total objective is to minimize:

$$\mathcal{L}_{\text{total}} = \mathcal{L}_{\text{TD}} + \lambda_1 \mathcal{L}_{\text{KL}} + \lambda_2 (\mathcal{L}_{\text{int}} + \mathcal{L}_{\text{inf}}) + \lambda_3 \mathcal{L}_{\text{DPP}}, \quad (12)$$

where $\mathcal{L}_{\text{TD}}$ represents the standard TD-error minimized via the Role-Augmented Mixer. The auxiliary terms shape the latent space: $\mathcal{L}_{\text{KL}}$ enforces the topological constraints of the GMM prior (Eq. 5); the functional terms $\mathcal{L}_{\text{int}}$ and $\mathcal{L}_{\text{inf}}$ (the latter combining $\mathcal{L}_{\text{rew}}$ and $\mathcal{L}_{\text{dyn}}$) ground the roles in specific intentions and their resulting influence; and $\mathcal{L}_{\text{DPP}}$ (Eq. 7) encourages diversity by minimizing the negative log-determinant of the kernel matrix. The coefficients $(\lambda_1, \lambda_2, \lambda_3)$ control the trade-off between these objectives.

## 5. Results

In this section, we conduct comprehensive experiments to validate the effectiveness and generalization capability of GTR. Specifically, our empirical evaluation is designed to answer the following research questions:

1. **Q1 (Generalization Analysis):** Does the learned role representation enable the policy to exhibit zero-shot and few-shot transfer capabilities in unseen scenarios?

2. **Q2 (Collaborative Efficiency):** Can GTR significantly improve collaborative efficiency and asymptotic performance through its role discovery in fixed tasks?

3. **Q3 (Ablation Analysis):** What are the contributions of the different components in GTR to the overall performance and generalization performance?

### 5.1. Experimental Setup

**Benchmarks.** We evaluate on StarCraft II benchmarks SMAC (Samvelyan et al., 2019) and SMACv2 (Ellis et al., 2023). While SMAC requires fine-grained coordination like

focus fire, SMACv2 introduces stochastic unit initialization to mitigate memorization. This setting compels agents to learn generalized behaviors and distinct divisions of labor, ensuring policies rely on adaptive tactics rather than fixed trajectories. To verify cross-task generalization, we select a subset of maps as source tasks and evaluate on unseen tasks.

**Baselines and Protocol.** We comprehensively evaluate GTR against state-of-the-art baselines. First, for zero-shot generalization scenarios, we select transfer-oriented methods that leverage specific inductive biases. These include entity-level approaches (ASN (Yang et al., 2023), UPDeT (Hu et al., 2021b), HPN (Jianye et al., 2022), RPG (Yao et al., 2025)) and the task-level bias method DT2GS (Tian et al., 2023). Second, in single-task settings, we evaluate our approach against representative structured algorithms. Specifically, we include role-based methods (RODE (Wang et al., 2020b), ROMA (Wang et al., 2020a), SIRD (Zeng et al., 2023), ACORM (Hu et al., 2023), R3DM (Goel et al., 2025)) and task-decomposition approaches (LSVQ (Zhou et al., 2024), LDSA (Yang et al., 2022)). Finally, we include the foundational method QMIX (Rashid et al., 2020) and QMIX-FT (Hu et al., 2021a) as general baselines. All experiments are conducted across four random seeds, reporting the mean and standard deviation of test win rates. Detailed hyperparameters are provided in Appendix D.

### 5.2. Cross-Task Zero-Shot and Few-Shot Performance

This section evaluates GTR in both zero-shot generalization and few-shot transfer scenarios. In the zero-shot setting, agents are required to solve unseen tasks using prior coordination experience without additional training. The few-shot setting tests the agents' ability to rapidly adapt to new tasks with limited samples, validating whether knowledge reuse facilitates positive transfer. Both scenarios present significant challenges due to dynamic changes in entity scales and action spaces. For the zero-shot scenario (Table 1), we compare GTR against transfer-oriented methods. Models are first trained on source tasks and then evaluated on a set of unseen target tasks. For the few-shot scenario (Figure 3), we fine-tune pre-trained models and compare the performance against training from scratch to assess transfer efficiency. Additionally, we provide supplementary generalization experiments on SMACv2 in Appendix E.1 to verify the effectiveness of our method on more challenging tasks.

**Zero-shot Generalization Analysis.** As presented in Table 1, GTR exhibits superior zero-shot generalization, significantly outperforming baselines across varying task scales and heterogeneity levels. Notably, while GTR may occasionally be slightly outperformed by specialized baselines on the source tasks (indicated by the light gray background), it demonstrates a much more pronounced and consistent performance advantage when transitioned to unseen target

*Table 1.* Zero-Shot Generalization Performance on SMAC. The rows with a light gray background represent the source tasks, while the others are unseen target tasks. **Bold** indicates the best performance.

| Task | ASN | UPDeT | DT2GS | HPN | RPG | GTR |
|------|-----|-------|-------|-----|-----|-----|
| 5m_vs_6m | $0.85 \pm 0.06$ | $0.83 \pm 0.04$ | $0.86 \pm 0.07$ | $\mathbf{0.95 \pm 0.08}$ | $0.93 \pm 0.05$ | $0.92 \pm 0.04$ |
| 7m | $0.28 \pm 0.16$ | $0.94 \pm 0.11$ | $0.97 \pm 0.02$ | $0.95 \pm 0.11$ | $1.00 \pm 0.00$ | $\mathbf{1.00 \pm 0.00}$ |
| 8m_vs_9m | $0.03 \pm 0.08$ | $0.14 \pm 0.16$ | $0.22 \pm 0.06$ | $0.15 \pm 0.15$ | $0.56 \pm 0.21$ | $\mathbf{0.91 \pm 0.04}$ |
| 10m_vs_11m | $0.00 \pm 0.00$ | $0.06 \pm 0.15$ | $0.14 \pm 0.06$ | $0.05 \pm 0.10$ | $0.72 \pm 0.24$ | $\mathbf{0.85 \pm 0.08}$ |
| 10m_vs_12m | $0.00 \pm 0.00$ | $0.00 \pm 0.00$ | $0.00 \pm 0.00$ | $0.00 \pm 0.00$ | $0.02 \pm 0.05$ | $\mathbf{0.10 \pm 0.02}$ |
| 3s_vs_5z | $0.88 \pm 0.09$ | $0.87 \pm 0.03$ | $0.97 \pm 0.02$ | $0.98 \pm 0.06$ | $0.97 \pm 0.06$ | $\mathbf{0.98 \pm 0.02}$ |
| 3s_vs_3z | $0.86 \pm 0.23$ | $0.98 \pm 0.08$ | $0.96 \pm 0.03$ | $0.86 \pm 0.22$ | $0.99 \pm 0.02$ | $\mathbf{0.99 \pm 0.01}$ |
| 3s_vs_4z | $0.85 \pm 0.16$ | $0.99 \pm 0.03$ | $0.99 \pm 0.02$ | $0.99 \pm 0.04$ | $0.98 \pm 0.04$ | $\mathbf{0.99 \pm 0.01}$ |
| 4s_vs_6z | $0.54 \pm 0.31$ | $0.93 \pm 0.09$ | $0.79 \pm 0.06$ | $0.90 \pm 0.12$ | $0.94 \pm 0.13$ | $\mathbf{0.97 \pm 0.02}$ |
| 4s_vs_7z | $0.23 \pm 0.26$ | $0.62 \pm 0.17$ | $0.10 \pm 0.04$ | $0.75 \pm 0.27$ | $0.58 \pm 0.31$ | $\mathbf{0.92 \pm 0.04}$ |
| 4s_vs_8z | $0.04 \pm 0.14$ | $0.23 \pm 0.12$ | $0.00 \pm 0.00$ | $0.00 \pm 0.00$ | $0.25 \pm 0.12$ | $\mathbf{0.72 \pm 0.09}$ |
| 3s5z_vs_3s6z | $0.84 \pm 0.08$ | $0.78 \pm 0.05$ | $0.91 \pm 0.05$ | $0.90 \pm 0.05$ | $\mathbf{0.93 \pm 0.03}$ | $0.91 \pm 0.04$ |
| 2s3z | $0.75 \pm 0.20$ | $0.02 \pm 0.04$ | $0.67 \pm 0.09$ | $0.48 \pm 0.25$ | $0.82 \pm 0.14$ | $\mathbf{0.84 \pm 0.05}$ |
| 3s5z | $0.96 \pm 0.09$ | $0.92 \pm 0.11$ | $0.94 \pm 0.03$ | $0.91 \pm 0.12$ | $0.96 \pm 0.04$ | $\mathbf{0.96 \pm 0.03}$ |
| 2s7z | $0.06 \pm 0.11$ | $0.36 \pm 0.16$ | $0.98 \pm 0.02$ | $0.28 \pm 0.18$ | $0.98 \pm 0.08$ | $\mathbf{0.98 \pm 0.02}$ |
| 4s5z_vs_4s7z | $0.00 \pm 0.00$ | $0.00 \pm 0.00$ | $0.17 \pm 0.06$ | $0.07 \pm 0.11$ | $0.18 \pm 0.25$ | $\mathbf{0.47 \pm 0.09}$ |

environments. We attribute this performance gap to the limitations of the inductive biases employed by baselines. First, methods that rely on entity-level inductive biases, such as ASN, UPDeT, HPN, and RPG, achieve permutation invariance via attention or hypernetworks but fail to capture the knowledge of collaborative logic among agents. Consequently, when team composition shifts significantly (e.g., from 3s5z_vs_3s6z to more complex 4s5z_vs_4s7z), these methods fail to adapt, lacking a semantic understanding of how such changes impact coordination. Second, DT2GS, based on task-level inductive biases, attempts generalization via explicit sub-task decomposition. However, this rigid decomposition struggles to adapt to changing environments where the optimal sub-task distribution shifts drastically. In contrast, GTR employs a role-level inductive bias. By extracting invariant role embeddings, GTR captures distinct functional divisions, which remain consistent regardless of agent size. This enables GTR to maintain robust performance even in complex heterogeneous tasks.

**Few-shot Transfer Analysis.** Figure 3 demonstrates GTR's efficacy in few-shot transfer. Compared to "scratch", which suffers from sample inefficiency and stagnates near zero in 4s5z_vs_4s7z, GTR achieves significantly faster convergence and higher asymptotic performance. Notably, in heterogeneous tasks with severe shifts, GTR exhibits a transient drop followed by a steep ascent. This signifies the correction of value overestimation and policy realignment. GTR's rapid recovery validates its robust role assignment. By leveraging the compositional generalization of the pre-learned role space, GTR effectively transforms the hard-exploration problem into a simpler task of recalibrating mixture weights.

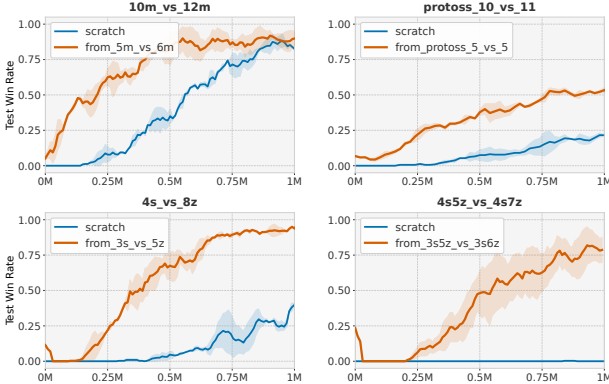

*Figure 3.* GTR's Few-shot performance on SMAC and SMACv2.

This allows the agent to bypass the bottleneck of learning from scratch, confirming that GTR successfully transfers structural knowledge to efficiently solve novel scenarios.

### 5.3. Fixed task Performance on SMAC and SMACv2

We conduct extensive experiments on SMAC and SMACv2 to validate the coordination efficiency and adaptability of GTR. As shown in Figure 4, GTR achieves state-of-the-art performance across most high-difficulty tasks. Additional experimental results on other maps are provided in Appendix E.2. In standard SMAC (columns 1-2), particularly on maps requiring intricate coordination such as 6h_vs_8z, GTR demonstrates significantly faster convergence and superior final win rates compared to baselines. This success underscores the efficacy of our structured role discovery in

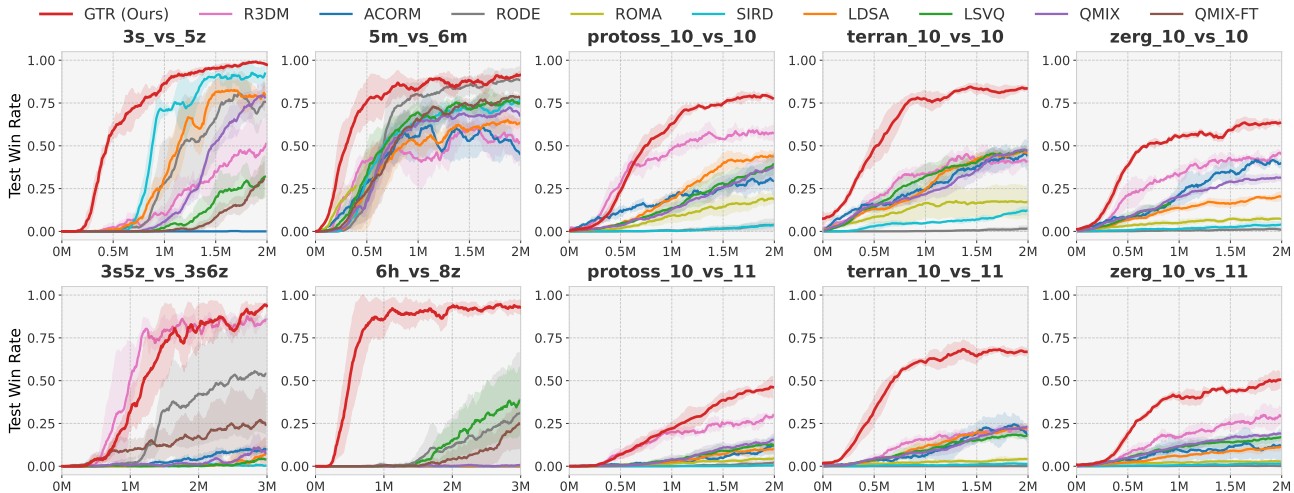

*Figure 4.* Performance comparison of GTR against baseline methods on SMAC (1-2 columns) and SMACv2 (3-5 columns) benchmarks. The solid lines represent the mean win rate across four random seeds, and the shaded regions indicate the standard deviation.

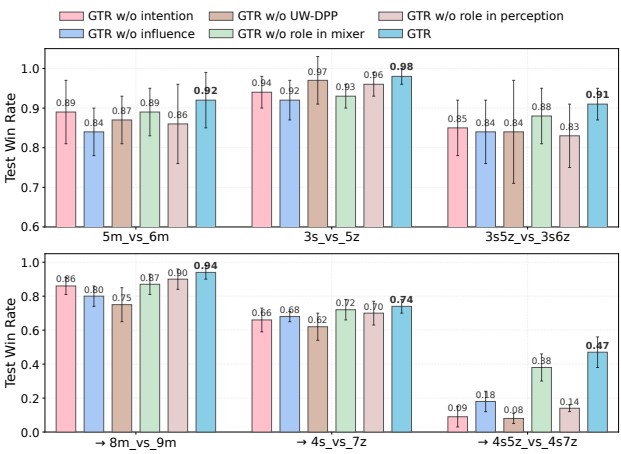

*Figure 5.* Ablation study of different GTR components.

organizing complex team behaviors.

SMACv2 (columns 3-5) introduces significant stochasticity by randomizing unit types and start positions. Unlike baselines that tend to memorize fixed trajectories, GTR's superior performance stems from its dynamic role adjustment. By modeling roles on structured latent distributions that incorporate agent intention and influence, this design endows the policy with the capability to cope with such stochasticity and adapt to varying team configurations. Furthermore, the agent network enhances perception via role embeddings, reinforcing robustness against the drastic initialization shifts characteristic of SMACv2.

In contrast, baselines exhibit distinct limitations in these settings. Methods lacking explicit role modeling, such as QMIX, struggle to achieve effective coordination in complex scenarios. Existing role-based methods also face challenges:

RODE and SIRD often overfit specific action sequences, leading to poor adaptation in highly stochastic tasks, while ROMA is hindered by ambiguous role boundaries. Although ACORM and R3DM show promise, they struggle to extract consistent role representations under stochastic conditions, with R3DM further limited by the high computational overhead of trajectory prediction. Finally, task-decomposition methods like LDSA and LSVQ fail to capture fine-grained coordination patterns in dynamic environments. Overall, GTR significantly enhances sample efficiency and stability through robust, transferable role discovery.

### 5.4. Ablation Results

Figure 5 presents the ablation results across both source and zero-shot scenarios. GTR consistently outperforms all variants, validating the necessity of each component. In source tasks (top row), intention and influence predicting contribute to training stability; however, their impact is significantly magnified in zero-shot settings (bottom row). This suggests that explicit action abstraction and dynamic modeling are crucial for filtering environment-specific noise, thereby facilitating adaptation to novel tasks.

Most notably, the removal of UW-DPP leads to a precipitous performance drop in unseen tasks, demonstrating that diverse role assignment is indispensable for preventing overfitting and ensuring robust transfer. Furthermore, the superiority of GTR over the variant lacking role-conditioned perception indicates that top-down perception (embedding role in the observation encoder) is far more critical for effective policy transfer than late integration.

Interestingly, removing roles from the mixer results in the smallest performance drop. This is consistent with

the CTDE paradigm, as zero-shot execution relies exclusively on decentralized policies, limiting the mixer's influence to the training phase. Furthermore, in Appendix E.4, we replace the GMM role prior with alternative distributions to verify the superiority in capturing diverse role features. We also conduct a sensitivity analysis on the number of roles to explore the impact of this hyperparameter on the model's overall performance and generalization ability in Appendix E.5. Collectively, these results confirm that GTR's generalization capability stems from the synergistic integration of diversity-guided role assignment, role-regularized dynamics, and top-down perception.

## 6. Conclusion

In this paper, we address the challenge of cross-task generalization in multi-agent reinforcement learning (MARL) by introducing a role-level inductive bias, which effectively mitigates the coordination and transfer difficulties arising from variations in team scales and compositions across tasks. Positioned as an intermediate abstraction between entity-level and task-level biases, this role-level inductive bias not only promotes policy diversity but also significantly enhances policy generalization through the inherent reusability and composability of roles. Specifically, we propose an adaptive role assignment mechanism that leverages a Gaussian Mixture Model (GMM) as the latent role space, integrated with a permutation-invariant role encoder and a UW-DPP mechanism to guarantee diversity in role assignment. Furthermore, an intention-influence decoder is introduced to constrain roles into learning transferable representations. Ultimately, these roles are embedded into the entire perception-decision pipeline to achieve efficient role utilization. Robust zero-shot and few-shot transfer results across tasks with varying scales and compositions demonstrate that our method enables efficient knowledge transfer and fast adaptation through the reusable role mechanism.

## 7. Limitations and Future Work

Although our approach yields superior generalization performance, it remains constrained by its sensitivity to diverse agent types. In particular, effective policy transfer is highly dependent on consistent agent types across source and target tasks. Since role semantics are tightly coupled with the inherent capabilities of agents, existing roles cannot be directly generalized to unseen agents with distinct attributes.

In future work, we aim to design an extensible role space to accommodate novel roles introduced by teams with new unit types. Building upon this, we will further explore knowledge retention mechanisms to effectively mitigate catastrophic forgetting during the expansion of the role space, thereby ensuring the stability of existing role knowledge.

## Acknowledgements

This work was supported by the National Natural Science Foundation of China (Grant No. 62576029).

## Impact Statement

Our work aims to advance Multi-Agent Reinforcement Learning (MARL). While the immediate application is within simulated environments, the broader implications concern the development of efficient and generalizable learning algorithms. We believe there are no specific ethical concerns or negative societal consequences to highlight.

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

# A. Theoretical Background: Permutation Invariance in Scalable MARL

In scalable Multi-Agent Reinforcement Learning (MARL), agents must generalize across tasks with varying numbers of interacting entities (e.g., scaling from $N = 5$ to $N = 20$ units). A critical challenge in this setting is that the dimension of the observation vector varies with the number of entities.

For instance, an agent's local observation $o_i$ is typically a concatenation of relative features for all other visible units (e.g., distance, health, type). If each entity possesses $d_f$ features (e.g., $d_f = 5$), the total observation dimension becomes $D_{obs} = d_f \times (N_{allies} + N_{enemies})$ (e.g., in SMAC). Standard feedforward networks (MLPs) require fixed input dimensions and thus cannot directly process such dynamic observations. To overcome this, PI-based methods reformulate the raw variable-length observation into an unordered set of entity embeddings $\mathcal{E} = \{e_1, \ldots, e_M\}$, which are then processed by permutation-invariant architectures to extract a fixed-dimensional representation.

## A.1. Formal Definitions: Invariance and Equivariance

Let the observation be decomposed into a set of $M$ entity feature vectors $\mathcal{E} \in \mathbb{R}^{M \times d_f}$, where $M$ varies across episodes or timesteps. The permutation group $S_M$ acts on $\mathcal{E}$ by reordering the entity sequence.

**Permutation Invariance (PI).** A function $f : \bigcup_{M \geq 1} \mathbb{R}^{M \times d_f} \to \mathbb{R}^{d_{out}}$ is permutation invariant if it maps any permutation of the input entity set to the same fixed-size output:

$$f(\pi \cdot \mathcal{E}) = f(\mathcal{E}), \quad \forall \pi \in S_M. \tag{13}$$

This property is crucial for aggregating the variable number of entity features into a unified *state representation* or *communication message* that remains stable regardless of the number of agents involved.

**Permutation Equivariance (PE).** A function $g : \mathbb{R}^{M \times d_f} \to \mathbb{R}^{M \times d'}$ is permutation equivariant if:

$$g(\pi \cdot \mathcal{E}) = \pi \cdot g(\mathcal{E}), \quad \forall \pi \in S_M. \tag{14}$$

PE is employed to extract entity-specific representations (e.g., attention weights or unique identifiers) while preserving the structural correspondence between the input entities and their processed features.

## A.2. Architectural Realizations for Variable-Dimension Observations

To process the entity set $\mathcal{E}$ (derived from the variable observation vector), we employ architectures that rely on shared weights and symmetric aggregation, decoupling the model parameters from the entity count $M$.

**Set Transformer and Pooling by Multihead Attention (PMA).** The Set Transformer (Lee et al., 2019) treats the observation as a set of interacting entities. It uses Multihead Attention (MHA) to capture pairwise interactions between entities:

$$\text{MHA}(Q, K, V) = \text{Concat}(H_1, \ldots, H_h)W^O, \quad \text{where } H_i = \text{Attention}(QW_i^Q, KW_i^K, VW_i^V). \tag{15}$$

To compress the variable-sized set $\mathcal{E}$ into a fixed-size representation $z$ (handling the variable dimension problem), we utilize *Pooling by Multihead Attention* (PMA). We introduce $k$ learnable seed vectors $S \in \mathbb{R}^{k \times d}$ to query the entity set:

$$z = \text{PMA}_k(\mathcal{E}) = \text{MHA}(S, \text{rFF}(\mathcal{E}), \text{rFF}(\mathcal{E})), \tag{16}$$

where rFF is a row-wise feedforward network applied to each entity embedding. This mechanism allows the agent to extract critical information from any number of surrounding entities into a fixed-size latent vector.

**DeepSets.** DeepSets (Zaheer et al., 2017) offers a computationally efficient baseline. It processes each entity feature vector $e_i$ independently using a shared encoder $\phi$ and aggregates them via a symmetric function $\rho$ (typically sum or mean):

$$z = \rho \left( \sum_{i=1}^{M} \phi(e_i) \right). \tag{17}$$

This structure explicitly handles the variable input dimension by summing over the entity axis, reducing the $M \times d_f$ input to a fixed $d_{out}$ vector.

**Graph Neural Networks (GNNs).** In scenarios where entities have explicit topological relationships (e.g., communication range), GNNs (Wu et al., 2020) generalize the set concept. The node features $h_i$ (initialized from entity observations $e_i$) are updated via Message Passing:

$$h_i^{(l+1)} = \text{Update}\left(h_i^{(l)}, \bigoplus_{j \in \mathcal{N}(i)} \text{Message}(h_i^{(l)}, h_j^{(l)})\right). \tag{18}$$

A global, symmetric *Readout* function then aggregates all node features into the fixed-size task embedding $z$. This allows the architecture to scale naturally to any number of nodes (agents/enemies) while capturing local interaction structures.

## B. Theoretical Background: Determinantal Point Processes

In this section, we provide a rigorous introduction to Determinantal Point Processes (DPPs) and detailed the formulation of the Utility-Weighted DPP (UW-DPP) utilized in our multi-agent role assignment.

### B.1. Standard Determinantal Point Processes

A Determinantal Point Process (DPP) is a probabilistic model defined over the power set $2^{\mathcal{Z}}$ of a discrete ground set $\mathcal{Z} = \{1, \dots, N\}$ (Kulesza et al., 2012). It elegantly captures negative correlations (repulsion) among items, making it an ideal tool for modeling diversity.

**Marginal Probabilities.** Strictly speaking, a DPP is defined by a marginal kernel matrix $\mathcal{K} \in \mathbb{R}^{N \times N}$ (where $0 \preceq \mathcal{K} \preceq I$). For any random subset $\mathbf{Y}$ drawn from the DPP, the probability that a subset $A \subseteq \mathcal{Z}$ is included in $\mathbf{Y}$ is given by the determinant of the marginal kernel restricted to $A$:

$$P(A \subseteq \mathbf{Y}) = \det(\mathcal{K}_A), \tag{19}$$

where $\mathcal{K}_A \equiv [\mathcal{K}_{ij}]_{i,j \in A}$.

**L-Ensembles.** In machine learning applications, it is often more convenient to model the atomic probabilities of observing a specific subset $S$ directly. This is achieved via L-ensembles. An L-ensemble defines the probability mass function using a symmetric positive semi-definite kernel matrix $L \in \mathbb{R}^{N \times N}$ (Tremblay et al., 2023):

$$P(\mathbf{Y} = S) = \frac{\det(L_S)}{\det(L + I)}, \tag{20}$$

where $I$ is the $N \times N$ identity matrix. The normalization constant follows from the identity $\sum_{S \subseteq \mathcal{Z}} \det(L_S) = \det(L + I)$.

**Relationship between $\mathcal{K}$ and $L$.** The marginal kernel $\mathcal{K}$ and the L-ensemble kernel $L$ provide alternative views of the same process and are related via the following matrix equation:

$$\mathcal{K} = L(L + I)^{-1} = I - (L + I)^{-1}. \tag{21}$$

The eigendecomposition of $L$ offers further insight: if $L$ has eigenvalues $\lambda_i$ and eigenvectors $\boldsymbol{v}_i$, then $\mathcal{K}$ shares the same eigenvectors but has eigenvalues $\frac{\lambda_i}{1+\lambda_i}$, representing the marginal probability of including the corresponding eigen-mode.

**Geometric Interpretation.** Let $L$ be the Gram matrix of feature vectors $B \in \mathbb{R}^{d \times N}$, such that $L_{ij} = \boldsymbol{b}_i^\top \boldsymbol{b}_j$. The determinant $\det(L_S)$ equals the squared volume of the parallelepiped spanned by the feature vectors in $S$:

$$\det(L_S) = \text{Vol}^2(\{\boldsymbol{b}_i\}_{i \in S}). \tag{22}$$

This geometric property drives diversity: if items in $S$ are similar (linearly dependent), the spanned volume collapses to zero, forcing $P(S) \to 0$.

### B.2. Limitations: The Blind Rejection Problem

Standard DPPs rely purely on geometric similarity (e.g., cosine similarity of embeddings) to determine exclusion. This creates a "blind rejection" issue in cooperative MARL:

- **Scenario**: Two agents $i$ and $j$ possess distinct high-utility roles but have similar embeddings ($z_i \approx z_j$).

- **Consequence**: The standard kernel yields $L_{ij} \approx 1$, causing $\det(L_{\{i,j\}}) \approx 0$.

- **Result**: The high-value pair is rejected solely due to representation redundancy, potentially harming team utility.

### B.3. Utility-Weighted Kernel Formulation

To address blind rejection, we construct a utility-weighted kernel that balances geometric diversity with explicit role utility. At timestep $t$, the kernel entries are defined as:

$$L_{ij} = b \cdot (u_i u_j)^2 \cdot \mathcal{K}(z_t^i, z_t^j), \tag{23}$$

where:

- $u_i \in \mathbb{R}^+$ is the utility score of role $i$.

- $\mathcal{K}(z_t^i, z_t^j) \in [-1, 1]$ measures the similarity between role latent embeddings.

- $b$ is a scalar controlling the determinant magnitude.

By decomposing the kernel into quality $q_i = \sqrt{b} u_i^2$ and similarity $S_{ij} = \mathcal{K}(z_t^i, z_t^j)$, we obtain $L_{ij} = q_i q_j S_{ij}$. The probability becomes proportional to $\left(\prod_{i \in S} u_i^2\right)^2 \det(S_S)$. This ensures that similar roles can co-exist if their combined utility mass is sufficient to counteract the determinantal penalty.

## C. Details of Mixing Network

In this section, we describe the implementation of the mixing network. To enable zero-shot transfer, we replace the fixed-size global state input with a permutation-invariant embedding generated via a two-stage attention mechanism.

### C.1. Attention-Based Global Representation

We construct the mixing weights by aggregating information from agents and other entities (e.g., enemies). Let $\mathbf{H}^{ag} = \{[s_{t,i}^{ag}, z_t^i]\}_{i=1}^N$ denote the role-augmented agent features and $\mathbf{H}^{oth} = \{s_{t,j}^{oth}\}_{j=1}^M$ denote the features of other entities.

First, we employ Multi-Head Self-Attention (MHA) to capture the cooperative relationships among agents, incorporating their role information:

$$\mathbf{C}^{ag} = \text{MHA}(\mathbf{Q} = \mathbf{H}^{ag}, \mathbf{K} = \mathbf{H}^{ag}, \mathbf{V} = \mathbf{H}^{ag}), \tag{24}$$

where the output $\mathbf{C}^{ag}$ represents the agent-group embedding.

Second, to capture the interaction between our team and the opponents, we use Multi-Head Cross-Attention (MHCA). Here, the agent features serve as queries to attend to the entities in the environment:

$$\mathbf{C}^{oth} = \text{MHCA}(\mathbf{Q} = \mathbf{H}^{ag}, \mathbf{K} = \mathbf{H}^{oth}, \mathbf{V} = \mathbf{H}^{oth}). \tag{25}$$

The final global embedding is obtained by aggregating these representations and passing them through an MLP, which then generates the mixing weights $W$ and biases $b$.

### C.2. Loss Function

The entire architecture is trained end-to-end. We minimize the Temporal-Difference (TD) error. Crucially, the joint action-value function $Q_{tot}$ is conditioned explicitly on the global state $s_t$ and the inferred role structure $\mathbf{z}_t$, ensuring the credit assignment adapts to the current role distribution:

$$\mathcal{L}_{TD} = \mathbb{E}_{(\boldsymbol{\tau}, \mathbf{a}, s, \mathbf{z}) \sim \mathcal{D}} \left[ \left( y^{tot} - Q_{tot}(\boldsymbol{\tau}_t, \mathbf{a}_t, s_t, \mathbf{z}_t; \theta) \right)^2 \right]. \tag{26}$$

Here, the target value is given by:

$$y^{tot} = r_t + \gamma \max_{\mathbf{a}'} Q_{tot}(\boldsymbol{\tau}_{t+1}, \mathbf{a}', s_{t+1}, \mathbf{z}_{t+1}; \theta^-), \tag{27}$$

where $\theta^-$ denotes the parameters of the target network, and $\mathbf{z}_t$ represents the set of role embeddings for all agents.

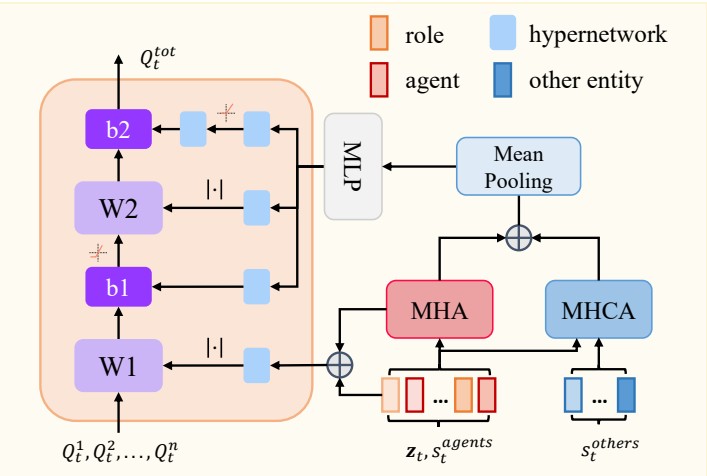

*Figure 6.* Overview of the GTR's Role-Based Mixer. Unlike conventional methods that rely on fixed-dimensional global states, GTR generates mixing weights via a permutation-invariant global embedding. (Right) The mixing representation is constructed by aggregating agent-specific features (reinforced by role embeddings $z_t$) and other entities (e.g., enemies) through Multi-Head Attention (MHA) and Multi-Head Cross-Attention (MHCA), respectively. (Left) We use aggregated features to generate the weights and biases for the mixing network, ensuring that the global value function $Q_{tot}$ captures the complex, role-dependent interactions while maintaining monotonicity.

## D. Hyperparameter Settings

We implement our GTR framework and the majority of baselines using the PyMARL[1] framework. For other baselines such as ACORM and R3DM, we utilize their respective original architectures and maintain the default hyperparameter settings as specified in their official implementations to ensure a fair comparison.

All experiments are conducted on a workstation equipped with an AMD Ryzen 9 9950X 16-Core Processor and an NVIDIA GeForce RTX 5090 GPU.

## E. Additional Experimental Results.

### E.1. Cross-Evaluation and Zero-Shot Generalization in SMACv2

Table 3 presents the cross-evaluation results. We benchmark our method against RPG, which is currently the state-of-the-art method utilizing entity-level inductive biases in SMAC. For all GTR experiments, we fixed the number of discoverable roles at $K = 4$ to evaluate the adaptability of a consistent role architecture.

**Trade-off between Small-Scale Fitting and Generalization.** In small-scale scenarios (e.g., source task 5_vs_5), RPG exhibits competitive performance, slightly outperforming GTR in specific in-distribution tests (e.g., Protoss: 0.73 vs. 0.70). This can be attributed to two factors: 1) The entity-level bias of RPG allows it to effectively overfit to the specific micro-patterns of small teams; 2) With $K = 4$, GTR might face slight over-parameterization when the number of agents is small (only 5), incurring a minor learning overhead.

However, this trade-off pays off in generalization. When transferring these policies to larger maps (10_vs_10), RPG suffers a significant performance drop (e.g., Protoss: $0.73 \rightarrow 0.54$), exposing its lack of scalability. In contrast, GTR generalizes significantly better (e.g., Protoss: $0.70 \rightarrow 0.65$; Terran: $0.73 \rightarrow 0.79$), proving that the learned roles capture transferable tactical semantics rather than memorizing fixed unit correlations.

**Dominance in Large-Scale Scenarios.** The advantages of GTR become overwhelming in larger source tasks (10_vs_10 and 10_vs_11), where the complexity of the state space increases. Unlike the small-scale setting, GTR consistently outperforms RPG in both in-distribution and cross-task metrics.

- **In-Distribution:** On the 10_vs_10 Terran task, GTR achieves a win rate of **0.83**, surpassing RPG's 0.82.

---
[1] https://github.com/oxwhirl/pymarl

*Table 2.* Detailed Hyperparameter Settings for GTR Framework.

| Category | Hyperparameter | Value |
|---|---|---|
| **GTR Specifics (Section 4)** | | |
| Role Space | Number of GMM Components ($K$) | 3 |
| | Latent Role Dimension ($d_z$) | 16 |
| Loss Weights | KL Weight ($\lambda_1$) | 0.01 |
| | Auxiliary Regularization Weight ($\lambda_2$) | 0.1 |
| | UW-DPP Weight ($\lambda_3$) | 0.05 |
| Regularization | Influence KL-div Weight ($\beta$) | 0.1 |
| | UW-DPP Perturbation ($\epsilon$) | $10^{-5}$ |
| | UW-DPP Scaling Base ($b$) | 3 |
| Encoder | Context Embedding Dim ($d_{\mathrm{model}}$) | 64 |
| | GRU Hidden State Dim | 64 |
| Decoder | Intention Branch Hidden Dim | 32 |
| | Influence Branch Hidden Dim | 32 |
| **Optimization & Training** | | |
| Optimization | Optimizer | Adam |
| | Learning Rate ($\alpha$) | $5 \times 10^{-4}$ |
| | Gradient Norm Clip | 10 |
| | Batch Size ($B$) | 32 |
| | Discount Factor ($\gamma$) | 0.99 |
| Exploration | Policy | $\epsilon$-Greedy |
| | Start $\epsilon$ | 1.0 |
| | End $\epsilon$ | 0.05 |
| Replay Buffer | Buffer Size | 5,000 |
| | Target Network Update Interval | 200 |
| **Zero-Shot Evaluation** | | |
| Test Settings | Evaluation Batch Size | 32 |
| | Episodes per Evaluation | 64 |

- **Robustness under Asymmetry:** When trained on 10_vs_10 and tested on the disadvantaged 10_vs_11 scenario, GTR maintains high resilience (e.g., Zerg: **0.40** vs. RPG's 0.37).

This indicates that while entity-level methods struggle to scale, GTR's role-based abstraction effectively handles complex, large-scale dynamics without sacrificing performance on the training task.

### E.2. Additional Results on SMAC and SMACv2

In addition to the main results, we evaluated GTR on a diverse set of maps from both SMAC and SMACv2 to further verify its robustness and generalization capabilities. Figure 7 illustrates the learning curves.

On the SMAC benchmark (top row), GTR exhibits remarkable sample efficiency. Notably, in the super-hard map MMM2 and the imbalance map 5z_vs_1ul, GTR achieves a high win rate significantly faster than strong baselines like R3DM and QMIX-FT. In the 1c3s8z_vs_1c3s9z scenario, GTR reaches a near-perfect win rate within just 0.25M steps, demonstrating its ability to identify effective strategies in asymmetrical combat quickly.

On the SMACv2 benchmark (bottom row), which introduces stochasticity in unit starting positions and types, GTR maintains a consistent lead. In protoss_5_vs_5 and terran_5_vs_5, our method outperforms the runner-up algorithms by a clear margin

*Table 3.* **Cross-Evaluation Matrix (Win Rate).** Comparison between baseline (RPG) and ours (GTR). The data is presented as *Mean* $\pm$ *Std*. **Bold** indicates the best performance. Underlining denotes in-distribution performance (Train Map = Test Map).

| Race | Source Task | Test on 5_vs_5 | | Test on 10_vs_10 | | Test on 10_vs_11 | |
|------|-------------|------|-----------|------|-----------|------|-----------|
| | | RPG | GTR(Ours) | RPG | GTR(Ours) | RPG | GTR(Ours) |
| **Zerg** | 5_vs_5 | **0.57 ± 0.07** | 0.57 ± 0.08 | 0.50 ± 0.13 | **0.54 ± 0.09** | 0.29 ± 0.07 | **0.34 ± 0.11** |
| | 10_vs_10 | 0.25 ± 0.08 | **0.33 ± 0.06** | 0.45 ± 0.06 | **0.62 ± 0.08** | 0.37 ± 0.09 | **0.40 ± 0.06** |
| | 10_vs_11 | 0.26 ± 0.08 | **0.41 ± 0.09** | 0.48 ± 0.09 | **0.60 ± 0.09** | 0.35 ± 0.07 | **0.56 ± 0.07** |
| **Protoss** | 5_vs_5 | **0.73 ± 0.09** | 0.70 ± 0.11 | 0.54 ± 0.10 | **0.65 ± 0.10** | 0.14 ± 0.05 | **0.18 ± 0.07** |
| | 10_vs_10 | 0.43 ± 0.13 | **0.51 ± 0.07** | 0.63 ± 0.11 | **0.80 ± 0.08** | 0.24 ± 0.07 | **0.32 ± 0.05** |
| | 10_vs_11 | 0.43 ± 0.08 | **0.46 ± 0.06** | 0.73 ± 0.08 | **0.86 ± 0.05** | 0.33 ± 0.09 | **0.53 ± 0.09** |
| **Terran** | 5_vs_5 | **0.76 ± 0.06** | 0.73 ± 0.08 | 0.70 ± 0.08 | **0.79 ± 0.06** | 0.42 ± 0.13 | **0.49 ± 0.10** |
| | 10_vs_10 | 0.55 ± 0.10 | **0.61 ± 0.05** | 0.82 ± 0.06 | **0.83 ± 0.05** | 0.53 ± 0.09 | **0.61 ± 0.07** |
| | 10_vs_11 | 0.48 ± 0.07 | **0.59 ± 0.11** | 0.80 ± 0.05 | **0.92 ± 0.05** | 0.56 ± 0.07 | **0.71 ± 0.07** |

in terms of final win rate. Even in the difficult zerg_5_vs_5 scenario, where most algorithms struggle to exceed a 50% win rate, GTR achieves the highest performance with lower variance. These results suggest that GTR's representation learning capability is highly effective in handling the dynamic and stochastic nature of complex multi-agent tasks.

### E.3. Extensibility Results Analysis of GTR

To evaluate the extensibility and universality of our proposed method, we integrated GTR into two representative value decomposition baselines: VDN and QMIX. To ensure a rigorous and fair comparison, we compared GTR against the fine-tuned versions of these baselines (denoted as VDN-FT and QMIX-FT). Crucially, to rule out the possibility that performance gains stem solely from increased model capacity, we scaled up the network size (i.e., hidden dimensions) of the base VDN-FT and QMIX-FT models. Figure 8 illustrates the performance comparison on the challenging 5m_vs_6m map in terms of Test Win Rate and Average Return.

**Consistent Performance Gains:** As shown in Figure 8, GTR consistently improves the performance of both baselines. For the non-linear decomposition method QMIX-FT, incorporating GTR improves the win rate from 0.83 to 0.91, demonstrating GTR's ability to further refine high-performing policies. More notably, for the linear decomposition method VDN-FT, GTR yields a substantial performance leap, boosting the win rate from 0.68 to 0.88 (an improvement of over 29%). A similar trend is observed. GTR+QMIX-FT achieves the highest average return of 19.2, surpassing the original QMIX-FT (18.2). Meanwhile, GTR+VDN-FT increases the return from 16.8 to 18.6.

The results validate GTR as an effective plug-and-play role discovery method for existing MARL baselines. The substantial performance enhancements observed on both QMIX-FT and VDN-FT stem from the improved team division of labor facilitated by our approach. By explicitly inferring latent roles and their relationships, GTR encourages agents to specialize and coordinate more efficiently.

### E.4. Ablation Study: Latent Space Priors

To validate the effectiveness of the proposed GMM-based role representation, we conducted an ablation study comparing our method against two representative baselines: (1) **Gaussian Distribution**, which utilizes a standard unimodal Gaussian prior (typical of VAEs), and (2) **Discrete Codebook**, which employs a categorical distribution (typical of VQ-VAEs). Table 4 presents the performance comparison on the source task (5m_vs_6m) and five unseen generalization scenarios.

As shown in the first row of Table 4, all three methods achieve comparable high performance on the training map 5m_vs_6m, with our GMM-based GTR achieving a slightly higher win rate ($0.92 \pm 0.04$). This indicates that introducing a structured GMM prior does not hinder the learning process; rather, it effectively captures the role distribution inherent in the asymmetric combat scenario.

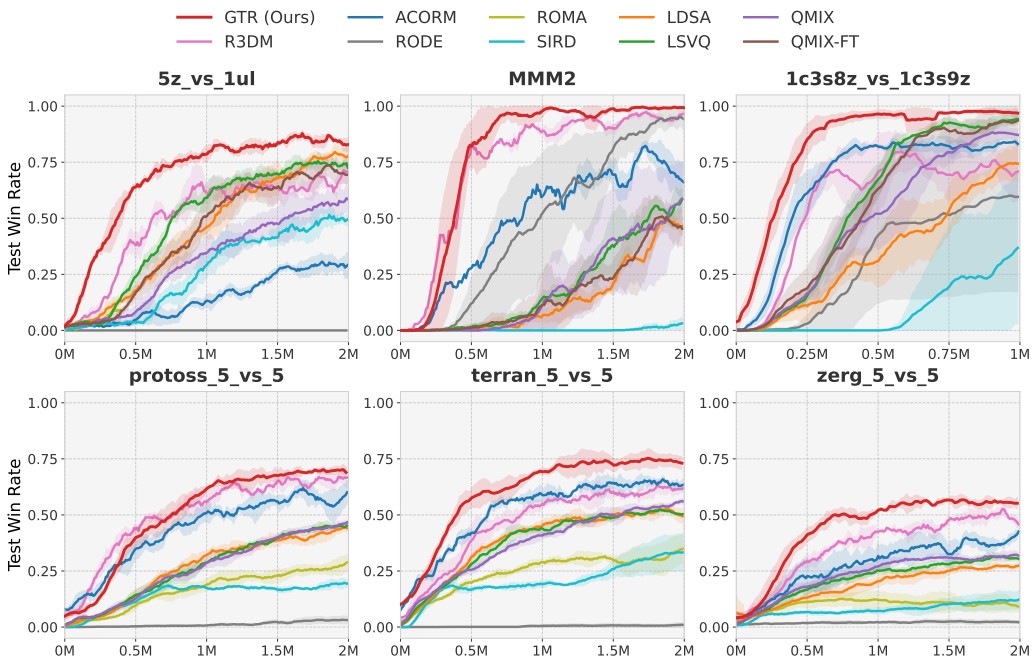

*Figure 7.* Performance comparison on additional SMAC and SMACv2 scenarios. The top row displays results on SMAC maps, while the bottom row shows results on SMACv2 scenarios.

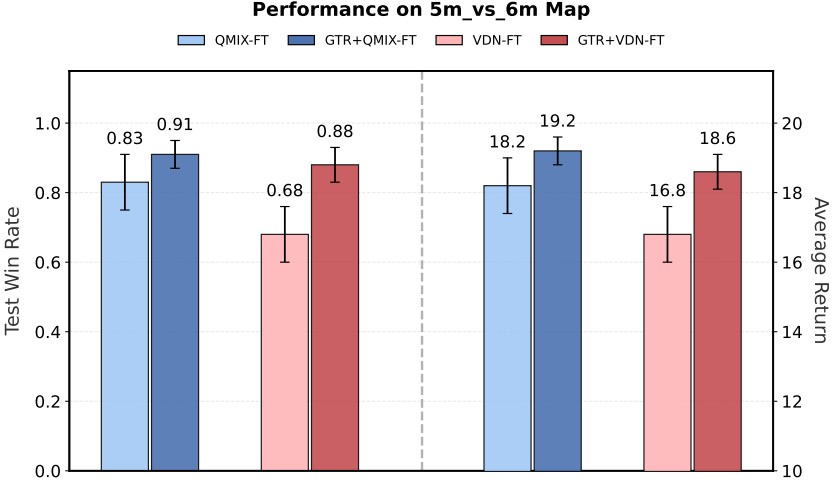

*Figure 8.* Performance improvements of GTR on fine-tuned QMIX-FT and VDN-FT.

The advantages of the GMM prior become evident in zero-shot transfer tasks, particularly as the scale and difficulty of the scenarios increase.

- **Comparison with Discrete Codebook (The Rigidity Problem):** The Discrete Codebook method shows significant degradation in large-scale scenarios. While it performs reasonably well on 3m, its performance collapses on 8m_vs_9m (0.61) and 10m_vs_11m (0.33). We attribute this to the rigidity of the discrete representations. The fixed codebook lacks the continuous flexibility required to adapt micro-management strategies when the number of agents doubles (from 5 to 10). Notably, on map 7m, the discrete baseline exhibits high instability (std $\pm 0.16$), whereas our method achieves a perfect win rate ($1.00 \pm 0.00$).

- **Comparison with Gaussian Distribution (The Over-smoothing Problem):** The Gaussian baseline maintains better stability than the discrete baseline but consistently underperforms our method across all unseen tasks. For instance, in 10m_vs_11m, the Gaussian prior lags behind our method by approximately $8\%$ (0.77 vs. 0.85). This supports our hypothesis that the unimodal Gaussian prior tends to over-smooth distinct tactical behaviors, leading to ambiguous role representations that fail to separate critical roles (e.g., specific focus-firing vs. kiting) in complex battles.

In the most challenging scenario, 10m_vs_12m, where agents face a significant numerical disadvantage, our method is the only one to retain a non-trivial win rate (0.10), outperforming the baselines, which drop to near zero.

These results confirm that the GMM prior strikes a crucial balance: it provides the structural discreteness needed to distinguish different roles (surpassing the Gaussian prior) while maintaining the intra-class variance required for policy adaptation (surpassing the Discrete Codebook).

*Table 4.* Performance Comparison on Source and Unseen Tasks. We report the mean test win rate $\pm$ standard deviation. Models are trained on 5m_vs_6m and evaluated on other maps in a zero-shot manner. Best results are highlighted in **bold**.

|  | Map | Gaussian Distribution | Discrete Codebook | GMM(GTR) |
|---|---|---|---|---|
| **Source Task** | 5m_vs_6m | $0.91 \pm 0.08$ | $0.89 \pm 0.06$ | $\mathbf{0.92 \pm 0.04}$ |
| **Unseen Tasks** | 3m | $0.88 \pm 0.04$ | $0.93 \pm 0.08$ | $\mathbf{0.97 \pm 0.03}$ |
|  | 7m | $0.96 \pm 0.02$ | $0.89 \pm 0.16$ | $\mathbf{1.00 \pm 0.00}$ |
|  | 8m_vs_9m | $0.86 \pm 0.06$ | $0.61 \pm 0.21$ | $\mathbf{0.91 \pm 0.04}$ |
|  | 10m_vs_11m | $0.77 \pm 0.06$ | $0.33 \pm 0.18$ | $\mathbf{0.85 \pm 0.08}$ |
|  | 10m_vs_12m | $0.04 \pm 0.02$ | $0.01 \pm 0.03$ | $\mathbf{0.10 \pm 0.02}$ |

### E.5. Ablation Study: Sensitivity to the Number of Roles

To determine the optimal granularity for the latent role space, we conducted an ablation study on the number of roles $K$, varying from 2 to 5, across three transfer scenarios with increasing complexity. Table 5 reports the win rates on both source tasks and unseen target tasks.

On the source tasks, we observe that the model requires a minimum level of expressiveness to handle complex interactions. While $K = 2$ performs adequately on simple homogeneous maps, it fails to capture the necessary tactical diversity in heterogeneous settings, achieving only a $0.56$ win rate on the *Mixed* source task compared to $0.91$ for $K = 3$. This significant gap suggests that a binary role space is insufficient to disentangle distinct behaviors such as tanking and kiting. However, performance saturates as $K$ reaches 3, with no meaningful gains observed for $K = 4$ or 5, indicating that the tactical primitives of the source maps are well-encapsulated by three abstract roles.

The critical trade-off between expressiveness and overfitting is revealed in the zero-shot transfer performance. We observe a clear inverted-U trend where generalization peaks at $K = 3$. In the *Marines* transfer scenario ($5m \rightarrow 8m$), $K = 3$ achieves a robust win rate of $\mathbf{0.90}$, whereas increasing $K$ further leads to severe degradation, dropping to $0.66$ with $K = 4$ and $0.57$ with $K = 5$. A similar collapse is seen in the *Mixed* scenario, where $K = 5$ performs poorly ($0.11$) compared to $K = 3$ ($\mathbf{0.36}$). This phenomenon suggests that a higher number of roles encourages the model to over-segment the agent group, encoding specific formations or local correlations unique to the source map's scale (e.g., specific positioning for 5 units). These over-fitted roles become brittle and fail to scale when applied to larger teams. Consequently, we identify $K = 3$ as the optimal hyperparameter, striking the best balance between learning diverse tactics and maintaining the abstraction level necessary for scale-invariant generalization.

*Table 5.* Ablation study on the number of roles ($K$) across three transfer scenarios. We report the Mean Win Rate on both **Source** and **Target** tasks. **Bold** indicates the best generalization performance. All source tasks are trained for $2M$ time steps.

| Roles ($K$) | Marines (Homogeneous) | | Stalkers (Kiting) | | Mixed (Heterogeneous) | |
|---|---|---|---|---|---|---|
| | 5m_vs_6m | → 8m_vs_9m | 3s_vs_5z | → 4s_vs_7z | 3s5z_vs_3s6z | → 4s5z_vs_4s7z |
| 2 | $0.86 \pm 0.07$ | $0.84 \pm 0.07$ | $0.95 \pm 0.05$ | $0.66 \pm 0.09$ | $0.56 \pm 0.15$ | $0.10 \pm 0.06$ |
| **3** | $\mathbf{0.92 \pm 0.07}$ | $\mathbf{0.90 \pm 0.08}$ | $\mathbf{0.98 \pm 0.02}$ | $\mathbf{0.92 \pm 0.07}$ | $\mathbf{0.91 \pm 0.04}$ | $\mathbf{0.36 \pm 0.08}$ |
| 4 | $0.91 \pm 0.03$ | $0.66 \pm 0.11$ | $0.97 \pm 0.03$ | $0.90 \pm 0.05$ | $0.85 \pm 0.05$ | $0.08 \pm 0.04$ |
| 5 | $0.90 \pm 0.05$ | $0.57 \pm 0.12$ | $0.92 \pm 0.07$ | $0.47 \pm 0.07$ | $0.79 \pm 0.10$ | $0.11 \pm 0.06$ |

### E.6. Visualization and Qualitative Analysis of Role-Based Attention

To demonstrate how the proposed GTR facilitates sophisticated micro-tactics among heterogeneous agents, we visualize the evolution of attention weights in the SMAC 2s3z scenario. In this map, agents comprise two distinct roles with complementary attributes:

- **Stalkers (rs, bs):** Ranged units characterized by high mobility and high damage output, ideal for "kiting" (hit-and-run) tactics but vulnerable to direct damage.

- **Zealots (rz, bz):** Melee units with high health points, serving as "tanks" to absorb damage and protect the rear line.

Figure 9 compares the attention distributions of GTR (Attention w/o Role) and GTR (Attention w/ Role) at Time Steps 5 (early engagement) and 10 (mid-game).

In heterogeneous combat, optimal strategy often requires prioritizing high-value, low-HP units (Stalkers) over durable units (Zealots).

**GTR (w/ Role):** The policy demonstrates learned tactical. At $t = 5$ (Right-Red matrix), the allied Stalker rs1 bypasses the nearest enemy tank bz1 and directs its maximal attention (0.41) toward the enemy Stalker bs1. Simultaneously, allied Zealots rz2 and rz3 converge fire on the same target bs1 (weights ≈ 0.40). This indicates that the role embedding successfully guides the agents to execute a "focus fire" strategy against the highest threat. At $t = 10$, after eliminating the primary threat, the team consistently switches focus to the remaining Zealot bz2, showing a unified strategic transition.

**GTR (w/o Role):** The method demonstrates a standard proximity-based strategy. At $t = 5$ (Left-Red matrix), rs1 focuses on bz1 (0.30), likely identifying it as the nearest immediate threat. While this is a valid policy, the attention weights are more uniformly distributed across the matrix compared to the role-based approach, indicating a less specialized focus-fire prioritization.

Intra-team attention reflects the agents' ability to maintain formation and protect vulnerable units.

**GTR (w/ Role):** The attention patterns reveal a "Tank-DPS" symbiotic relationship. At $t = 5$ (Right-Blue matrix), the ranged unit rs1 pays high attention to the allied tank rz1 (0.41), while rz1 reciprocally attends to rs1 (0.40). This suggests rs1 is positioning itself relative to its protector, while rz1 is actively peeling for the damage dealer.

**GTR (w/o Role):** The attention appears more diffuse across the team. For instance, rs1 maintains a moderate attention weight (0.33) on rz2 while also attending to other allies. This pattern suggests that without explicit role guidance, the agents adopt a strategy of global team awareness, monitoring the aggregate state of allies rather than forming the sharp, pairwise structural bonds seen in the role-based method.

The visualization confirms that the inclusion of role embeddings transforms the agents' behavior from stochastic reactions to structured, role-aware tactics. By explicitly modeling agent types, GTR reduces the search space for policy learning, enabling the emergence of advanced behaviors such as focus firing on key targets and cooperative defensive formations.

## F. Algorithm Pseudocode

In this section, we provide the detailed training procedure for the GTR framework. Algorithm 1 outlines the end-to-end optimization process, including the permutation-invariant encoding, structured variational role inference, and the

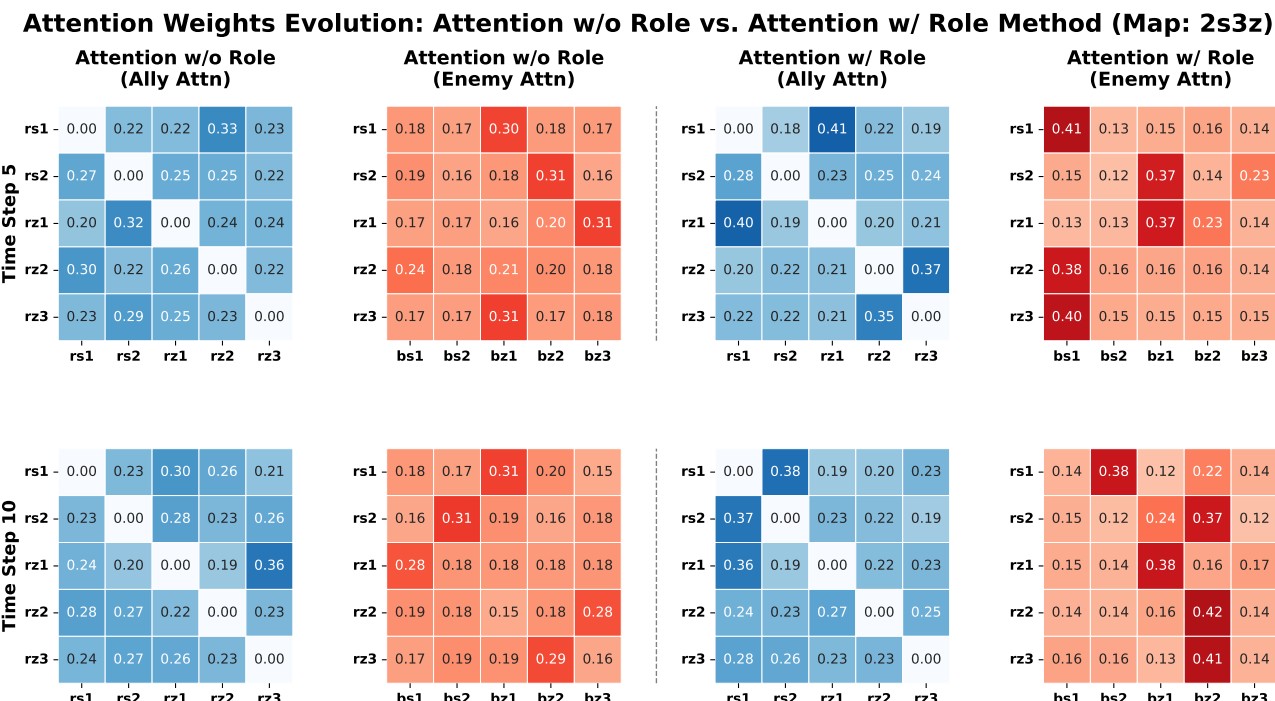

*Figure 9.* Visualization of attention weight evolution on the 2s3z map. The heatmaps compare the attention distribution of GTR (w/o Role) and GTR (w/ Role) at Time Step 5 and 10. The map consists of heterogeneous agents: Stalkers (ranged units: red stalker (rs), blue stalker (bs)) and Zealots (melee units: red zealots (rz), blue zealots (bz)). Right (w/ Role): Exhibits clear tactical patterns. Ranged Stalkers (rs) focus on high-threat enemy Stalkers (bs) rather than the nearest tanks (bz), demonstrating intelligent target selection. Left (w/o Role): Shows scattered attention, where agents fail to differentiate between unit types, leading to suboptimal focus fire. (Note: Darker colors indicate higher attention weights. "Ally Attn" masks self-attention.)

computation of auxiliary regularization objectives (UW-DPP, Intention, and Influence).

---

**Algorithm 1** GTR: Gaussian-Mixture-Model-Based Transferable Roles Discovery

---

1: **Input:** Team size $N$, Replay Buffer $\mathcal{D}$, Batch size $B$
2: **Input:** GMM components $K$, Hyperparameters $\lambda_1, \lambda_2, \lambda_3, \beta, \epsilon$
3: **Initialize:** Encoder $q_\phi$, Policy $\pi_\theta$, Decoders $\{D_{\text{int}}, D_{\text{dyn}}, R_{\text{rew}}\}$, Mixer Net
4: **Initialize:** GMM Priors $\boldsymbol{\mu}_k, \boldsymbol{\Sigma}_k$ for $k = 1 \dots K$
5: ─────────────────────────────────────────
6: **while** not converged **do**
7:    *// Phase 1: Interaction & Role-Conditioned Execution*
8:    **for** $t = 0$ to $T$ **do**
9:      **for** agent $i = 1$ to $N$ **do**
10:       Obtain observation $o_t^i$ and previous action $a_{t-1}^i$
11:       $x_t^i, c_t^i, z_t^i, u_t^i \leftarrow$ RoleInference$(o_t^i, a_{t-1}^i)$       *// Infer role latent*
12:       $\mathbf{q}_i \leftarrow$ RoleConditionedQuery$(\mathbf{e}_{\text{ego}}^i, c_t^i, z_t^i)$       *// Section 4.3*
13:       Select action $a_t^i \sim \pi_\theta(\cdot|o_t^i, c_t^i, z_t^i)$
14:      **end for**
15:      Execute $\mathbf{a}_t$, receive $r_t, \mathbf{o}_{t+1}$
16:      Store transition $(\mathbf{o}_t, \mathbf{a}_t, r_t, \mathbf{o}_{t+1})$ in $\mathcal{D}$
17:    **end for**
18:    *// Phase 2: Joint Optimization*
19:    Sample random batch of trajectories $\mathcal{T} \sim \mathcal{D}$
20:    **Compute Role Losses:**
21:    **for** each timestep $t$ in batch **do**
22:      Extract PI embeddings $x_t$ and posteriors $q_\phi(z_t|x_t)$
23:      *1. Structured Variational Inference:*
24:      $\mathcal{L}_{\text{KL}} \leftarrow \text{KL}(q_\phi(c|x)\|p(c)) + \sum_k q(k)\,\text{KL}(q_\phi(z|k)\|\mathcal{N}(\boldsymbol{\mu}_k, \boldsymbol{\Sigma}_k))$
25:      *2. Geometric Coverage via UW-DPP:*
26:      Construct kernel $\mathbf{L}_{ij} = b \cdot (u_i u_j)^2 \cdot k(z_i, z_j)$ where $u_i = \max_k q_\phi(c = k|x_i)$
27:      $\mathcal{L}_{\text{DPP}} \leftarrow -\frac{1}{N} \log \det(\mathbf{L} + \epsilon \mathbf{I})$
28:      *3. Intention & Influence Streams:*
29:      Get action primitive $y_t = \mathcal{M}(a_t)$
30:      $\mathcal{L}_{\text{int}} \leftarrow \mathbb{E}_{q_\phi}\left[-\log p(\hat{a}_t|z_t)\right]$
31:      Predict next params $(\hat{\boldsymbol{\mu}}_{t+1}, \hat{\boldsymbol{\sigma}}_{t+1}) = f_{\text{dyn}}(z_t, a_t)$ and reward $\hat{r}_t$
32:      $\mathcal{L}_{\text{inf}} \leftarrow \mathbb{E}_{q_\phi}\left[D_{\text{KL}}\big(\text{sg}[q_\phi(z_{t+1}|x_{t+1})] \,\|\, \hat{p}(z_{t+1}|z_t, a_t)\big)\right] + \|r_t - f_{\text{rew}}(\bar{z}_t, \mathbf{a}_t)\|^2$
33:    **end for**
34:    **Compute RL Loss:**
35:    $Q_{tot} \leftarrow$ Mixer$(\mathbf{Q}_{local}, s_t, \mathbf{z}_t)$      *// Role-Aware Credit Assignment*
36:    $\mathcal{L}_{\text{TD}} \leftarrow (y_{target} - Q_{tot})^2$
37:    **Total Objective:**
38:    $\mathcal{L}_{\text{total}} \leftarrow \mathcal{L}_{\text{TD}} + \lambda_1 \mathcal{L}_{\text{KL}} + \lambda_2(\mathcal{L}_{\text{int}} + \mathcal{L}_{\text{inf}}) + \lambda_3 \mathcal{L}_{\text{DPP}}$
39:    Update parameters $\theta, \phi$ via gradient descent on $\mathcal{L}_{\text{total}}$
40: **end while**

---

---

**Algorithm 2** Sub-routine: Role Inference

---

1: **Input:** Local observation $o$, Previous action $a_{\text{prev}}$
2: **Output:** PI Embedding $x$, Role Embedding $z$, Confidence $u$
3: **Function** ROLEINFERENCE($o, a_{\text{prev}}$):
4:     *1. PI Encoding*:
5:     Parse $o$ into Ego $\mathbf{q}$ and Entities $\mathbf{K}, \mathbf{V}$
6:     $x \leftarrow \text{LayerNorm}\left(\mathbf{q} + \text{Attention}(\mathbf{q}, \mathbf{K}, \mathbf{V})\right)$
7:     *2. GMM Posterior Estimation*:
8:     Compute category posterior $q_\phi(c|x) = \text{softmax}(f_c(x))$
9:     Compute component parameters $\mu_\phi(x, k), \sigma_\phi(x, k)$ for $k \in \{1, \ldots, K\}$
10:    *3. Sampling*:
11:    Sample $k \sim q_\phi(c|x)$                                         *// Categorical*
12:    Sample $z \sim \mathcal{N}\left(\mu_\phi(x, k), \sigma_\phi(x, k)\right)$             *// Reparameterization*
13:    Compute Role Confidence $u = \max_k q_\phi(c = k|x)$
14:    **Return** PI Embedding $x$, Role Embedding $z$, Confidence $u$

---

