# OpenReview forum: "Role-Level Inductive Bias for Cross-Task Generalization in Multi-Agent Reinforcement Learning"
_ICML.cc/2026/Conference — ICML 2026 regular_

### Official Review · Reviewer_X6z2 · 2026-03-03

**Soundness:** 4
**Presentation:** 4
**Significance:** 3
**Originality:** 3
**Overall Recommendation:** 5
**Confidence:** 5

**Summary:**

This paper studies cross-task generalization in multi-agent reinforcement learning (MARL) and introduces role-level inductive bias as an intermediate abstraction between entity-level permutation invariance and task-level structural decomposition. The authors propose GTR, which uses a GMM-based latent role space, a Utility-Weighted DPP for diverse role assignment, and dual-stream regularization (intention + influence) for role decoupling. Experiments on SMAC and SMACv2 demonstrate strong zero-shot and few-shot transfer performance.

**Compliance With Llm Reviewing Policy:**

Affirmed.

**Key Questions For Authors:**

1. Multiple modules increase design and training overhead, relevant evaluation on it would be helpful.
2. In Table 1, the authors report the zero-shot generalization performance of different methods. However, how do these methods perform on the source tasks? In other words, if some baselines already perform worse than GTR on the original training tasks, then inferior zero-shot generalization results alone do not necessarily demonstrate weaker generalization ability. A comparison that accounts for source-task performance would provide a more convincing assessment of cross-task generalization.

**Limitations:**

yes

**Strengths And Weaknesses:**

**Strengths**
1. Cross-task generalization remains a central challenge in multi-agent reinforcement learning (MARL), and this paper addresses a important problem.
2. The idea of combining role discovery with cross-task generalization is well motivated. Introducing a *role-level* intermediate abstraction between entity-level and task-level inductive biases is conceptually clean and provides a compelling explanation for the limitations of existing paradigms.
3. The experimental evaluation is reasonably comprehensive. The comparisons include entity-level, role-based, and task-level baselines. The results consistently favor the proposed method, particularly under transfer settings.
4. The paper is well organized and clearly written, and the figures are of high quality.

 **Weaknesses**
1. The introduction of GMM inference, DPP-based selection, and dual-stream regularization increases computational overhead. Moreover, the presence of multiple modules makes it difficult to identify which component is the primary contributor to the performance gains.
2. The paper claims that the learned roles capture transferable tactical semantics. Providing visualization analyses of the learned role embeddings would offer stronger empirical evidence to substantiate this claim.

---

> ### Author Rebuttal · Authors · 2026-03-30
>
> Thank you for taking the time to review our paper and for recognizing the advantages of GTR in terms of structural clarity and experimental validation. We hope the following response will alleviate your concerns:
>
>
> ### To W1 & Q1 Computational overhead and module contribution
> We appreciate the reviewer's concern regarding overhead and contribution.
>
> **Module Contribution.** Beyond Figure 5, Appendix E.4 evaluates alternative designs. Replacing the GMM prior with a single Gaussian or discrete codebook (VQ-VAE) degrades performance, as GMM's soft clustering uniquely captures complex role distributions for generalization. We further compared UW-DPP against entropy and dual-stream against observation reconstruction (Table 1). Results show a clear hierarchy: GMM is the primary driver; UW-DPP and dual-stream are necessary complements addressing its failure modes (mode collapse and task-specific overfitting, respectively).
>
> | | GTR | Role Entropy (w/o UW-DPP) | Observation Reconstruction (w/o intention & influence) |
> | :--- | :--- | :--- | :--- |
> | 5m_vs_6m (src) | 0.92 $\pm$ 0.04 | 0.87 $\pm$ 0.05 | 0.87 $\pm$ 0.07 |
> | 10m_vs_11m (tgt) | 0.85 $\pm$ 0.08 | 0.76 $\pm$ 0.06 | 0.71 $\pm$ 0.04 |
>
> **Computational Overhead.** GTR's time cost is 1.71x that of QMIX, far lower than ROMA (3.13x), while delivering superior generalization. Crucially, GTR's transferability amortizes this cost: zero-shot deployment eliminates retraining, and few-shot adaptation converges faster than learning from scratch, yielding net savings in multi-task scenarios.
>
> ||Training Time(h)|GPU(GB)|Relative Time|
> |-|-|-|-|
> |QMIX [3]|2.4|0.75|1.00x|
> |GTR|4.1|1.63|1.71x|
> |ROMA [4]|7.5|0.96|3.13x|
>
> ### To W2 Role visualization
> To validate the tactical semantics of the learned roles, we conducted a [t-SNE visualization of the role embeddings](https://anonymous.4open.science/r/Rebuttal-8877/smac_tsne.png). The results clearly reveal distinct clustering patterns, where different agent role representations are mapped to separate and well-defined regions within the latent space. This topological separation provides compelling evidence that GTR successfully abstracts meaningful tactical roles. Furthermore, we have added detailed visualizations of agent role behaviors:
> 1. [Role behavior visualization in 5m_vs_6m](https://anonymous.4open.science/r/Rebuttal-8877/role_behavior_5m6m.pdf).
> 2. [Zero-shot transfer role behavior visualization (from 5m_vs_6m to the unseen 8m_vs_9m)](https://anonymous.4open.science/r/Rebuttal-8877/role_behavior_8m9m.pdf).
>
> The role-switching patterns presented in these figures directly demonstrate that GTR successfully learns generalizable tactical policies. Moreover, the distinct action preferences exhibited by each role highlight a clear functional behavioral differentiation at the tactical level.
>
>
> ### To Q2 Performance comparison on the source task
> As suggested, since source task performance is crucial for evaluating zero-shot generalization, we list these results for all methods below. While baselines achieve competitive source-task results, they suffer varying performance drops during zero-shot transfer. In stark contrast, GTR maintains a consistently high win rate, especially in complex scenarios (as corroborated by Table 1 in the paper). This clearly shows that GTR's superior generalization inherently stems from its transferable role abstraction, not just a stronger source-task policy.
>
>
> | Source task | ASN | UPDeT | DT2GS | HPN | RPG | GTR |
> | :--- | :--- | :--- | :--- | :--- | :--- | :--- |
> | 5m_vs_6m | 0.85 ± 0.06 | 0.83 ± 0.04 | 0.86 ± 0.07 | 0.95 ± 0.08 | 0.93 ± 0.05 | 0.92 ± 0.04 |
> | 3s_vs_5z | 0.88 ± 0.09 | 0.87 ± 0.03 | 0.97 ± 0.02 | 0.98 ± 0.06 | 0.97 ± 0.06 | 0.98 ± 0.02 |
> | 3s5z_vs_3s6z | 0.84 ± 0.08 | 0.78 ± 0.05 | 0.91 ± 0.05 | 0.90 ± 0.05 | 0.93 ± 0.03 | 0.91 ± 0.04 |
>
>
> Thank you again for your feedback, which will help us improve our paper.

---

> > ### Author Rebuttal · Reviewer_X6z2 · 2026-04-01
> >
> > The authors have addressed my concerns satisfactorily in the rebuttal, and I appreciate the clarifications.
> >
> > For further improving the paper’s impact and reproducibility, I encourage the authors to consider releasing the code upon publication.

---

> > > ### Author Response · Authors · 2026-04-01
> > >
> > > Thank you very much for your positive feedback and for confirming that our rebuttal has satisfactorily addressed your concerns. We highly appreciate your constructive suggestions on code release to improve the impact and reproducibility of our work. We will make our code publicly available. Thank you again for your valuable comments.

---

### Official Review · Reviewer_F8sw · 2026-03-03

**Soundness:** 3
**Presentation:** 3
**Significance:** 2
**Originality:** 2
**Overall Recommendation:** 3
**Confidence:** 5

**Summary:**

This paper addresses the challenge of cross-task generalisation in multi-agent reinforcement learning by introducing **role-level inductive bias**. This establishes an intermediate abstraction layer between entity-level permutation invariance and task-level sub-task decomposition. Based on this idea, the authors propose the **GTR** method: a Gaussian mixture model prior learns a structured role space; utility-weighted DPP ensures diversity and stability in role allocation; and roles are regularised through intention and impact objectives, abstracting them from environment-specific details. The learned role information is synchronously injected into both policy learning and credit assignment mechanisms, thereby supporting transfer learning across team sizes. Experiments on the SMAC and SMACv2 datasets demonstrate that, compared to existing methods, this model achieves improved performance when transferring to unseen tasks under both zero-shot and few-shot conditions.

**Compliance With Llm Reviewing Policy:**

Affirmed.

**Final Justification:**

After considering the paper and the rebuttal, my final recommendation remains **Weak Reject**: the work is technically solid and the rebuttal improved clarity on scope and design, but I am still not fully convinced that the proposed “role-level inductive bias” is conceptually distinct enough from prior role-discovery methods, and the evidence for broad cross-task generalization remains somewhat limited beyond the current benchmark family.

**Key Questions For Authors:**

1. What is the strict boundary of **role-level inductive bias**?

2. How extensive is your **zero-shot or few-shot cross-task generalization** in practice?

3. Are GMM, UW-DPP and intention(influence) each necessary and are they better than simpler alternatives?


**If you address my concerns, I will consider raising my score.**

**Limitations:**

While the paper acknowledges limitations (namely sensitivity to agent types and transfer restricted to compatible agent definitions), it should expand this with concrete failure cases and a clearer discussion of potential negative impacts and mitigation under distribution shift.

**Strengths And Weaknesses:**

### **Strengths**
1. The paper presents a well-structured architecture with a clear design
2. The experiments include ablations that clarify how each component affects performance.

### **Weaknesses**

1. The paper clasifies inductive bias into entity-, role-, task-level categories, positioning *role-level* as its core conceptual contribution; However, extensive prior work exists on role discovery/emergent roles (such as  ROMA,RODE,ACORM,R3DM). The paper's *role-level* approach primarily repackages classic role discovery objectives ( labour, diversity and collaboration) within a new hierarchical framework. It lacks a rigorously defined boundary and explanation of essential differences, so it may be regarded as incremental engineering rather than a novel concept or paradigm.
2. The paper claims to achieve superior *zero-shot* or *few-shot* transfer on unseen tasks, yet experiments primarily involve map or scale transfer within SMAC/SMACv2, falling under **intra-environmental transfer** within the same domain family. More critically, the conclusions and limitations **explicitly acknowledge** the method's sensitivity to agent types and that transferability is constrained by *compatible agent definitions*. This weakens the credibility of **cross-task generalisation** as a broad capability, suggesting it may be closer to benchmark-family transfer than more general cross-domain generalisation.
3.  The paper defines $u_i$ as the maximum role posterior probability (role confidence, range [0,1]) within UW-DPP, yet in the DPP background and appendix it treats $u_i$ as a general utility score $\mathbb{R}^+$, creating semantic and domain inconsistencies; Furthermore, symbol reuse and conflicts exist (such as $K$ denotes both the number of roles and the kernel function notation), making it difficult for readers to establish consistency between actual implementation and derivation.
4. GTR is composed of multiple stacked modules (such as GMM structural prior + KL, UW-DPP, multi-task regularisation for intention/influence, role-conditioned perception and mixer). Although ablation studies are present, these primarily show **performance degradation upon removal** without direct comparisons to alternative designs (such as  replacing DPP with entropy and coverage regularisers, substituting GMM with VQ or codebook or Dirichlet or single Gaussian models; or replacing intention/influence with more general self-supervised objectives). Without such controls, reviewers may interpret the contribution as merely a **complex combination enhancement** rather than a fundamental breakthrough driven by a specific novel mechanism.

---

> ### Author Rebuttal · Authors · 2026-03-30
>
> We sincerely appreciate your recognition and will adopt your helpful suggestions. Our responses are as follows:
> ### To W1 & Q1 The Strict Boundary of Role-Level Inductive Bias
> Thanks for this insightful question. We clarify the boundary between role-level inductive bias and prior role discovery.
>
> **The boundary is the design goal: fitting a single task vs. transferring across tasks.** Prior methods discover roles as behavioral clusters for within-task coordination. Their role spaces become entangled with task-specific details (fixed observation dimensions, specific action spaces) and collapse when tasks change. This is precisely because role discovery alone is necessary but not sufficient — it qualifies as role-level inductive bias only when stronger constraints are met: (1) roles are decoupled from task-specific observations to remain valid across tasks, (2) role diversity is maintained across varying team scales, and (3) role semantics capture transferable collaborative knowledge rather than environment-specific behaviors. As a concrete instantiation, GTR satisfies all three through GMM, UW-DPP, and intention/influence regularization.
>
> In essence, role discovery answers "how to diversify agent behaviors within a task," while role-level inductive bias builds upon this and further answers "how to make the learned division of labor reusable for coordinating agents in unseen tasks." Our zero-shot results empirically confirm this. We will add a formal discussion in the revision.
>
> ### To W2 & Q2 GTR's Cross-Task Generalization
> Thank you for this observation. Our contribution targets cross-task generalization under varying team scales and compositions, not cross-domain transfer, consistent with our title. We highlight two points. First, in-domain transfer is non-trivial: altered compositions drastically change joint state-action spaces, agent counts, type ratios, and tactical demands. Transferring from 3s5z_vs_3s6z to 4s5z_vs_4s7z requires genuine policy restructuring where most baselines fail (Table 1), and SMACv2's stochastic initialization prevents memorization, demanding adaptive coordination. Second, we discussed agent-type constraints for transparency. Cross-type transfer remains an open challenge as observation-action semantic mismatches are unsolved by current MARL methods. We will address this via type-agnostic agent roles in future work and revise the manuscript to scope our claims accordingly.
>
> ### To W3 Symbol Definition Problem
> We sincerely apologize for the confusion caused by the inconsistent mathematical notation and are deeply grateful for your meticulous review. In the revised manuscript and the Appendix, we will strictly standardize all mathematical symbols. To resolve the specific issue of notation reuse, we will consistently use $K$ to denote the number of roles and $\mathcal{K}$ for the kernel function. We will also perform a comprehensive audit of the entire text to ensure all other notations are consistent and well-defined.
>
> ### To W4 & Q3 Component effectiveness and substitutability
> We appreciate your concern. GTR is a concrete instantiation of role-level inductive bias, not a parallel stacking of independent modules. All components serve one goal: constructing a transferable role space where roles act as task-agnostic coordination primitives. Even when individual components are replaced with alternatives, the framework retains generalization (see table below and Appendix E.4) — the role-level bias still functions.
>
> However, each design is necessary to fully realize this bias, addressing a distinct failure mode: (1) **Why GMM?** Transfer requires roles both categorically distinct and behaviorally continuous. A single Gaussian collapses diversity; VQ-VAE loses intra-role variation. GMM captures complex distributions with smooth transitions (Appendix E.4). (2) **Why UW-DPP?** GMM is prone to mode collapse. Unlike entropy, which only encourages uniformity, UW-DPP jointly models inter-role repulsion and cooperative utility. (3) **Why intention/influence?** Roles must encode task-agnostic semantics. Observation reconstruction causes memorization of task-specific details, degrading transfer. In summary, role-level inductive bias provides foundational generalization, while each component resolves a specific bottleneck to realize it.
>
> | |GTR|Role Entropy (w/o UW-DPP)|Observation Reconstruction (w/o intention & influence)|
> |:---|:---|:---|:---|
> |5m_vs_6m(src)|0.92 $\pm$ 0.04|0.87 $\pm$ 0.05|0.87 $\pm$ 0.07|
> |10m_vs_11m(tgt)|0.85 $\pm$ 0.08|0.76 $\pm$ 0.06|0.71 $\pm$ 0.04|
>
>
> ### To Limitations
> We provide cross-agent failure cases. Homogeneous roles learn focus-fire but struggle with Stalker-Zealot kiting, showing role semantics link to agent capabilities. We will integrate continual learning into GTR, enabling the role space to progressively acquire unseen roles without forgetting prior coordination.
>
> || 2s3z(tgt) | 3s_vs_5z(tgt) |
> |- | - | - |
> | 5m_vs_6m(src) | 0.08 $\pm$ 0.04 | 0.11 $\pm$ 0.05 |

---

> > ### Author Rebuttal · Reviewer_F8sw · 2026-04-03
> >
> > Thanks for the detailed rebuttal. The added discussion on the role-level inductive bias, the clarification of the paper’s scope, and the extra analysis on the different components are helpful and address part of my concerns.
> >
> > I do think the rebuttal makes the paper clearer, especially on the motivation and design choices. That said, I still have some reservations about how clearly the conceptual novelty is separated from prior role-discovery work, and I think this point should be made more explicit in the final version. Overall, my concerns are only partially resolved.
> >
> > **Could the authors state more clearly what fundamentally distinguishes their role-level inductive bias from prior role-discovery methods, beyond improved transfer performance?**

---

> > > ### Author Response · Authors · 2026-04-05
> > >
> > > Thank you for the insightful follow-up. We appreciate the opportunity to clarify the conceptual novelty of our work.
> > >
> > > **Fundamentally, traditional role discovery learns environment-bound representations by passively overfitting to local observation-action trajectories, whereas our inductive bias enforces a proactive structural prior that constrains roles into task-agnostic coordination patterns while ensuring policy diversity. In other words, prior methods rigidly memorize the current task, whereas the role-level inductive bias ensures that roles not only adapt to the immediate task but also guide team collaboration in novel scenarios.**
> > >
> > > GTR explicitly restricts the optimization space, rather than allowing role representations to freely fit task-specific rewards and state transitions. Through the following series of constraints, we guide the role learning process to ensure that gradients only update within a task-agnostic, modular space.
> > >
> > > - **Intention over Action:** Task-specific action details are structurally blocked from the role representation. The network only encodes abstract intent, preventing roles from memorizing specific, non-transferable action mappings.
> > > - **Abstract Dynamics (Influence):** Roles predict role-to-role transitions and reward mappings entirely within the latent space, rather than from raw observation spaces.
> > > - **Structured Latent Geometry:** The GMM prior establishes a structured discrete–continuous role space, while UW-DPP enforces scale-invariant diversity. This geometric regularization ensures the space adapts to any team size, avoiding task-specific collapse.
> > >
> > > Without these structural constraints, traditional methods freely encode observation-action correlations. Retraining them yields an incompatible role space. In contrast, GTR not only learns diverse roles within a single task, but its structural constraints also guarantee that the learned roles remain consistent across different tasks.
> > >
> > > We systematically contrast these conceptual differences below (to be integrated into the final version):
> > >
> > > | Comparison Dimension | Traditional Role Discovery Methods | Role-Level Inductive Bias (GTR) |
> > > | :--- | :--- | :--- |
> > > | **Nature of Role Representation** | **Passive Descriptive Fitting:** Observation-action trajectory clustering. It learns representations tightly bound to specific local observations or state dynamics. | **Structurally Constrained Abstraction:** Cross-task abstraction of collaborative functions. GTR's roles explicitly separate semantic intent from task-specific execution. |
> > > | **Core Objective** | Optimizing the division of labor and sample efficiency within a single, fixed environment. | Extracting composable, task-agnostic policy priors that remain semantically invariant under distribution shifts. |
> > > | **Characteristics of Roles** | Task-specific, non-composable, and structurally bound to the training agent ID and team scale. | Task-agnostic, modular, composable, and intrinsically permutation-invariant. |
> > >
> > > We will make this distinction explicit in the introduction and related work of the final version. We sincerely appreciate your evaluation, which helped us sharply articulate our core contribution.

---

### Official Review · Reviewer_rpuK · 2026-03-05

**Soundness:** 3
**Presentation:** 3
**Significance:** 3
**Originality:** 3
**Overall Recommendation:** 5
**Confidence:** 4

**Summary:**

The authors investigate inductive biases in MARL generalization, particularly multi-agent collaboration in StarCraft II, proposing a role-level abstraction. The authors present a GMM-VAE-based framework where roles function less as semantic abstractions and more as dynamic meta-control signals. The method is evaluated on SMAC and SMACv2 benchmarks, demonstrating strong zero-shot and few-shot transfer performance over competitive baselines.

**Compliance With Llm Reviewing Policy:**

Affirmed.

**Final Justification:**

The paper is well motivated. The demonstrated phenomenon are interesting. Although I am not confident in my assessment of the method, I lean towards clear accept.

**Key Questions For Authors:**

It would be interesting if the authors could share more about the emergent behaviors under UW-DPP.

**Limitations:**

Yes.

**Strengths And Weaknesses:**

**Strength**
- The applicaiton of VAE-GMM is genuinely novel
- Few-shot transfer results are compelling.
- The UW-DPP formulation addresses a well-documented failure mode of standard DPPs ("blind repulsion") in a principled way.

** Weaknesses**
- Evaluation is entirely on StarCraft II
- The paper provides no systematic analysis of emergent role semantics. The single attention visualization in Figure 9 is suggestive but insufficient. Since this is a study solely on games MARL, I expect more detailed analysis.

I am not familiar with this line of work so my evaluation of novelty might be biased.

---

> ### Author Rebuttal · Authors · 2026-03-30
>
> We sincerely appreciate the reviewer's thorough evaluation and insightful feedback on our manuscript. In what follows, we carefully address your concerns and questions regarding our methodology.
>
> ### To W1 Additional Experimental Task
> To address your concern regarding our experimental evaluation, we conducted additional experiments on Level-Based Foraging (LBF) [1], a fully cooperative, non-adversarial resource collection task.  As demonstrated in the table below, GTR consistently achieves strong cross-task generalization. Furthermore, its performance surpasses the RPG baseline not only in the training scenario but also when generalizing to new grid sizes (10x10) and entity distributions (4p-3f), underscoring the effectiveness and structural advantages of our role discovery mechanism. We will add these LBF-related experimental results and the corresponding training details to the revised manuscript.
>
> ||15x15-3p-4f(src)|10x10-3p-4f(tgt)|15x15-4p-3f(tgt)|
> |---|---|---|---|
> |RPG [2]|0.56±0.03|0.61±0.04|0.68±0.03|
> |GTR|0.59±0.05|0.63±0.02|0.71±0.03|
>
> ### To W2 Analysis of Visualization Results
>
> Regarding your concern about role semantic analysis, we have provided comprehensive visualizations to address this.
>
> On the one hand, we visualized [agent role and behavior transitions](https://anonymous.4open.science/r/Rebuttal-8877/role_behavior_5m6m.pdf) during an episode in the 5m_vs_6m scenario. The action distribution per role confirms clear semantic separation: Role 0 almost exclusively executes "Move" actions, Role 1 heavily focuses on "Attack" actions, and Role 2 performs a balanced mix of both. The role transition charts further illustrate that agents dynamically adapt their roles at both micro and macro levels in response to the evolving state of the environment, such as team casualties.
>
> On the other hand, by [comparing game replays alongside these role transitions](https://anonymous.4open.science/r/Rebuttal-8877/smac_vis.pdf), we can clearly observe that agents execute precise tactics aligned with their assigned roles. For instance, at t=1, agents adopt the "Position" role to safely approach enemies; at t=6, they switch to "DPS" to perform focus fire; and at t=10, agents under heavy attack seamlessly transition to "Hit & Run" to kite enemies and mitigate damage. This shows that our framework is able to discover highly interpretable and tactically meaningful semantic roles.
>
> ### To Q1 The Emergent Behaviors under UW-DPP.
> We sincerely thank the reviewer for this interesting question. UW-DPP yields emergent behaviors by coupling diversity pressure to agent's role confidence. (1) Spontaneous specialization: in early training, uncertain role assignments cause the kernel to penalize similarity, pushing agents toward distinct behaviors. As shown in Appendix E.6 (Figure 9), Stalkers (ranged, high-damage) and Zealots (melee, durable) form tactical pairings (attention ~0.41). (2) Coordinated focus fire: as roles stabilize, utility weighting relaxes this pressure, allowing overlap. Agents jointly target high-threat enemies over nearby tanks. (3) Scale-adaptive transfer: $\det(\mathbf{L}_t)$ scales with team size, guiding new agents into unoccupied role regions. Ablation (Figure 5) confirms removing UW-DPP causes the largest transfer drop.
>
> [1] Papoudakis G, Christianos F, Schäfer L, et al. Benchmarking multi-agent deep reinforcement learning algorithms in cooperative tasks[J]. arXiv preprint arXiv:2006.07869, 2020.
>
> [2] Yao C, Lin Y, Song S, et al. From general relation patterns to task-specific decision-making in continual multi-agent coordination[J]. arXiv preprint arXiv:2507.06004, 2025.

---

> > ### Author Rebuttal · Reviewer_rpuK · 2026-03-31
> >
> > I greatly appreciate the analysis. I hope seeing a more complete version in the final version (thus only marked partially resolved). I have raised my OA.

---

> > > ### Author Response · Authors · 2026-04-01
> > >
> > > We sincerely appreciate your positive feedback. We fully accept your suggestion and will thoroughly supplement the relevant content to present a more complete and comprehensive version in the final manuscript. We will carefully revise this part to fully address your concern. Thank you again for your valuable comments.

---

### Official Review · Reviewer_G56g · 2026-03-09

**Soundness:** 4
**Presentation:** 3
**Significance:** 3
**Originality:** 2
**Overall Recommendation:** 4
**Confidence:** 4

**Summary:**

The paper proposes GTR, a framework for achieving cross-task generalization in Multi-Agent Reinforcement Learning (MARL). The authors' core idea is to introduce a role-level inductive bias as an intermediate abstraction lying between entity-level (permutation invariance) and task-level (subtask decomposition) prior approaches. GTR uses a GMM-structured latent space, a novel Utility-Weighted Determinantal Point Process (UW-DPP) for diverse role assignment, and role regularization via intention and  influence to produce transferable role representations. The authors evaluate GTR in a variety of SMAC and SMAC-v2 cross-task zero-shot and few-shot scenarios, achieving improved performance compared to a suite of baselines.

**Compliance With Llm Reviewing Policy:**

Affirmed.

**Key Questions For Authors:**

- The authors mention that actions are mapped to 3 logical modes: Idle, self-directed, and interaction (Lines 248-250). Could the authors clarify how the atomic actions are mapped to these modes in practice, and whether this mapping is manually specified or automatically determined?
- Could the authors include an analysis on training time and efficiency of GTR and how it compares to the baseline methods? There is minimal discussion of computational cost, and given that GTR adds substantial complexity on top of QMIX and similar foundational methods, it would help to contextualize the additional overhead.
- The authors use a sensitivity analysis to justify their selection of $K=3$ as the optimal number of Gaussian components for their GMM. However, the strength of the authors' claim about role granularity would be strengthened if they provided additional analyses into the structure of the latent space. Could  the authors provide (a) a PCA/T-SNE visualization of a representative latent role space and the location of the Gaussian components, (b) a BIC score analysis over different number of clusters to assess how well the Gaussian components cover the role space, (c) an analysis of how the roles differ in terms of action distributions/trajectories, enabling better intuition as to the types of behavioral differences that comprise the role space, and (d) a cross-task role correspondence analysis showing whether roles discovered in source tasks map semantically to roles in target tasks, which would directly verify the paper's central claim that roles constitute transferable collaborative knowledge.
- Could the authors provide a more comprehensive discussion in the related work section? Including the above-mentioned literature would help contextualize GTR better among adjacent MARL topics and literature.

**Limitations:**

Yes

**Strengths And Weaknesses:**

## Strengths
1. Task transferability in MARL is an important problem, particularly that of varying team sizes and compositions. The paper motivates this problem well and articulates the shortcomings of current PI and decomposition methods and why they fail to scale in these scenarios.
2. The authors evaluate GTR over a large set of baselines, spanning foundational methods like QMIX to more modern SOTA approaches like RPG and ACORM. The depth of baselines and large set of transfer task evaluations is commendable and is effective in contextualizing GTR with respect to modern approaches.
3. The ablation study of different GTR components is highly informative. Testing in both zero-shot and source scenarios is comprehensive and helpful to identify the effects of the various components of GTR, and the finding that removing UW-DPP leads to the sharpest performance drop supports the paper's argument that diverse role assignment is a crucial consideration for these types of MARL methods.

## Weaknesses
- The authors include a thorough discussion on MARL methods that deal use PI and task decomposition for transferability. However, there are some papers in the Ad-Hoc Teaming and Zero-Shot Coordination literature that propose methods that optimize zero-shot adaptation to novel _teammates_ instead of _tasks_, but have considerable conceptual and algorithmic overlap to GTR. MeLIBA [1] uses Bayesian partner modeling to maintain beliefs over partner type, which can transfer to unseen partners at test-time. TALENTS [2] and Hong et al [3] both learn latent role/strategy spaces from offline data, then train role-conditioned cooperators that recognize and adapt to new partners at test-time. TALENTS [2] and MESH [4], in addition, cluster their learned latent space to partition a discrete number of roles, similar to how GTR leverages its GMM prior. Lastly, Unsupervised Environment Design (UED) methods like PAIRED [5], ADD [6], and CEC [7], should also be mentioned, as they solve a similar problem to GTR of cross-task generalization but approach it from a different angle. Omission of these papers from the related work section limits the clarity of the paper with respect to the greater zero-shot and role inference MARL landscape.
- The paper only includes experiments in the SMAC/SMACv2 domains, which limits the argument of generalizability to other environments and domains. Experiments in other domains, particularly ones which much larger team sizes (e.g multi-robot swarm navigation) could help examine how GTR scales to much larger team sizes.
- The paper does not include substantive behavioral analysis of the discovered roles. Despite the claims about role transferability and disambiguation, there lacks an analysis showing what the $K$ different roles look like semantically. The attention visualization in Fig. 9 shows the effect of role conditioning on tactical focus, but further analysis about what distinguishes each role on an action-distribution or latent cluster level would greatly aid the intuition of GTR (for reference, the ROMA paper [8] includes a graphic which intuitively demonstrates the different types of agent roles over the course of an episode) .
[1] Zintgraf, Luisa, et al. "Deep interactive bayesian reinforcement learning via meta-learning." _arXiv preprint arXiv:2101.03864_ (2021).
[2] Li, Benjamin, et al. "Adaptively Coordinating with Novel Partners via Learned Latent Strategies." _arXiv preprint arXiv:2511.12754_ (2025).
[3] Hong, Joey, Sergey Levine, and Anca Dragan. "Learning to influence human behavior with offline reinforcement learning." _Advances in neural information processing systems_ 36 (2023): 36094-36105.
[4] Zhao, Michelle, Reid Simmons, and Henny Admoni. "Coordination with humans via strategy matching." _2022 IEEE/RSJ International Conference on Intelligent Robots and Systems (IROS)_. IEEE, 2022.
[5] Dennis, Michael, et al. "Emergent complexity and zero-shot transfer via unsupervised environment design." _Advances in neural information processing systems_ 33 (2020): 13049-13061.
[6] Chung, Hojun, et al. "Adversarial environment design via regret-guided diffusion models." _Advances in Neural Information Processing Systems_ 37 (2024): 63715-63746.
[7] Jha, Kunal, et al. "Cross-environment cooperation enables zero-shot multi-agent coordination." _arXiv preprint arXiv:2504.12714_ (2025).
[8] Wang, Tonghan, et al. "Roma: Multi-agent reinforcement learning with emergent roles." _arXiv preprint arXiv:2003.08039_ (2020).

---

> ### Author Rebuttal · Authors · 2026-03-30
>
> We appreciate your feedback and recognition. Here are our responses:
>
> ### To W1 & Q4 Supplementary Related Work
> Thank you for highlighting these works, which we will discuss in our revised Related Work. AHT/ZSC methods generalize along the teammate policy dimension, adapting zero-shot to novel partners. UED methods generalize along the environment dimension, dynamically generating diverse open-ended environments for policy generalization. In contrast, GTR generalizes along the cross-task dimension, uniquely leveraging abstract roles as transferable knowledge to bridge structural differences in varying team configurations.
>
> ### To W2 Additional Experimental Environment
> We evaluated GTR on Level-Based Foraging (LBF) [1], demonstrating zero-shot generalization. This proves that using role discovery to enhance cross-task generalization is not limited to SMAC, but transfers to other environments. Additionally, we extended experiments to larger SMAC maps. Future work will apply GTR to complex scenarios such as multi-robot navigation.
>
> ||15x15-3p-4f(src)|10x10-3p-4f(tgt)|15x15-4p-3f(tgt)|
> |-|-|-|-|
> |RPG [2]|0.56±0.03|0.61±0.04|0.68±0.03|
> |GTR|0.59±0.05|0.63±0.02|0.71±0.03|
>
> ||30m|corridor|
> |-|-|-|
> |RPG [2]|0.89±0.07|0.84±0.03|
> |GTR|0.94±0.06|0.87±0.05|
>
> ### To W3 & Q3 Role Behavior and Visualization Analysis
> We provide visualizations to illustrate GTR's behavior and role distribution in 5m_vs_6m.
>
> **[Role Behavior Visualization](https://anonymous.4open.science/r/Rebuttal-8877/role_behavior_5m6m.pdf):** The figure illustrates the role switching and the evolutionary trend of roles at the team level. This indicates that agents can adaptively adjust their policies to changing battlefield conditions, fostering coordinated team behaviors. Furthermore, the behavioral distributions vary significantly across different roles, verifying the semantic distinctiveness and behavioral consistency of the discovered roles. Additionally, we supplemented a [comparative visualization](https://anonymous.4open.science/r/Rebuttal-8877/smac_vis.pdf) between replay and the corresponding role transitions to provide intuitive evidence of this alignment.
>
> **[Role Clustering Visualization](https://anonymous.4open.science/r/Rebuttal-8877/smac_tsne.png):** The figure presents the t-SNE clustering of roles. This demonstrates that GTR can effectively distinguish different roles in the latent space, forming clear and well-separated clusters.
>
> **BIC Analysis:** The Table shows the Bayesian Information Criterion (BIC) values for different K. When K = 3, the BIC reaches its optimal (minimum) point, achieving the best trade-off between model complexity and goodness-of-fit. This justifies setting K = 3 for this task.
>
> |K|bic_gmm(1e6)|bic_all(1e6)|
> |---|---|---|
> |K=2|574.8|575.3|
> |K=3|534.1 ✓|534.7 ✓|
> |K=4|551.6|552.2|
> |K=5|558.0|558.5|
>
> **[Cross-Task Role Analysis](https://anonymous.4open.science/r/Rebuttal-8877/role_behavior_8m9m.pdf):** To elucidate GTR’s zero-shot transfer, we analyzed role behavioral distributions when generalizing from 5m_vs_6m to 8m_vs_9m. Results confirm learned roles project onto the target task, preserving semantic characteristics (similar action distributions), which explains GTR's generalization.
>
> ### To Q1 Action Mapping
> Thank you for this question. The mapping $\mathcal{M}$ is manually specified by design — a lightweight scheme applicable to most MARL domains with minimal domain knowledge: any agent action either does nothing, modifies own state, or affects another entity. This coarse-grained scheme enables roles to capture transferable functional intent rather than overfitting to specific action spaces, which would harm generalization. In SMAC: *Idle*=\{no-op, stop\}, *Self-Directed*=\{move[dir]\}, *Interaction*=\{attack[target]\}. We will gladly include a more detailed description in the revised manuscript.
>
>
> ### To Q2 Training Cost Assessment
> As shown below, GTR incurs a 71% time overhead compared to QMIX on an RTX 5090 but remains significantly faster than ROMA. This extra cost is well-justified by GTR’s faster convergence and superior zero-shot generalization.
>
> ||Training Time(h)|GPU(GB)|Relative Time|
> |-|-|-|-|
> |QMIX [3]|2.4|0.75|1.00x|
> |GTR|4.1|1.63|1.71x|
> |ROMA [4]|7.5|0.96|3.13x|
>
> [1] Papoudakis G, Christianos F, Schäfer L, et al. Benchmarking multi-agent deep reinforcement learning algorithms in cooperative tasks[J]. arXiv preprint arXiv:2006.07869, 2020.
>
> [2] Yao C, Lin Y, Song S, et al. From general relation patterns to task-specific decision-making in continual multi-agent coordination[J]. arXiv preprint arXiv:2507.06004, 2025.
>
> [3] Rashid T, Samvelyan M, De Witt C S, et al. Monotonic value function factorisation for deep multi-agent reinforcement learning[J]. Journal of Machine Learning Research, 2020, 21(178): 1-51.
>
> [4] Wang T, Dong H, Lesser V, et al. Roma: Multi-agent reinforcement learning with emergent roles[J]. arXiv preprint arXiv:2003.08039, 2020.

---

> > ### Author Rebuttal · Reviewer_G56g · 2026-04-03
> >
> > The authors did a good job of addressing the weaknesses and questions that we raised in the review, they were able to clarify most of the outstanding concerns we had regarding their submission.
> > In particular, the role space visualization was helpful to understand how the different roles are disambiguated in the latent space.
> > The minor remaining criticisms I have are as follows: (1) providing a similar semantic role analysis in the LBF environment to provide further context as to the roles that emerge in various environments, further strengthening the generalizability claim, and (2) describing the behavior of the 3 SMAC roles more clearly in text, the action distribution plots help but it's still a bit difficult to understand the difference between the 3 roles on a semantic level, especially between DPS and hit-and-run.
> >
> >
> > Overall, I would be happy to raise the rating to 5, as I think this paper is quite thorough and the authors did a good job of addressing our concerns in the rebuttal.

---

> > > ### Author Response · Authors · 2026-04-04
> > >
> > > We sincerely thank the reviewer for the positive re-evaluation. We are glad the role visualization was helpful. Below we address the two remaining points.
> > >
> > > **(1) Semantic Role Analysis in LBF**
> > >
> > > Based on [the Gantt chart (lower panel of the figure)](https://anonymous.4open.science/api/repo/Rebuttal-8877/file/lbf.pdf?v=b7990ba1) and [supplementary GIF animations](https://anonymous.4open.science/r/Rebuttal-8877/lbf-vis.gif), we conduct a semantic role analysis of **LBF-2s-10x10-3p-4f-v3**. GTR discovers three distinct roles forming a cooperative foraging pipeline:
> > >
> > > - **Scout (Blue):** Governs exploration. Agents execute broad sweeping trajectories prioritizing spatial coverage. The action distribution is dominated by directional moves with negligible LOAD attempts.
> > >
> > > - **Supporter (Orange):** Once food is detected, agents shift to goal-directed movement. The key distinction from the Scout role is intentionality—trajectories converge toward a target. The distance-to-food curve decreases monotonically.
> > >
> > > - **Collector (Green):** Tightly coupled with reward acquisition, correlating with LOAD actions and reward timestamps. This role activates only when joint-action preconditions are met, indicating that GTR has learned the cooperative constraint. Critically, the Collector role exhibits cooperative waiting, lingering near high-level food until a capable partner arrives, reflecting awareness of heterogeneous capabilities.
> > >
> > > The Scout-Supporter-Collector progression forms a complete search-approach-harvest cycle, paralleling the Position-DPS-Hit-and-Run progression in SMAC, jointly validating GTR's role semantics.
> > >
> > > **(2) Clarifying DPS vs. Hit-and-Run in SMAC**
> > >
> > > We provide a sharper semantic distinction for the three SMAC roles:
> > >
> > > - **Position (Blue):** Active during the opening phase ($t=0$–$4$), handling pre-engagement maneuvering to form favorable formations. The action profile is nearly 100% movement with zero attacks.
> > >
> > > - **DPS (Orange):** The sustained-damage role. Agents commit to continuous attacking with minimal repositioning (Attack:Move $\approx$ 4:1), maximizing damage through persistent focus fire, accepting positional risk for higher uptime.
> > >
> > > - **Hit-and-Run (Green):** The survival-aware damage role. Unlike DPS, agents interleave attacks with deliberate retreats (Attack:Move $\approx$ 2.3:1). The critical difference is *damage avoidance*: agents kite—attacking then withdrawing before retaliation—trading throughput for longevity.
> > >
> > > The DPS vs. Hit-and-Run distinction mirrors the classic *maximize output* vs. *survive to keep outputting* trade-off. GTR dynamically switches agents between these roles based on real-time conditions, sustaining effective combat under attrition.
> > >
> > > We hope these analyses fully address the remaining concerns. We believe the enriched evidence across both LBF and SMAC further strengthens the paper, and would be grateful if the reviewer could consider reflecting this in the final rating.

---

### Decision · Program_Chairs · 2026-04-30

**Decision:**

Accept (regular)

**Comment:**

I recommend weak accept. The paper presents a technically solid approach to cross-task MARL transfer, with strong empirical results, informative ablations, and a rebuttal that resolved many reviewer questions about efficiency, source-task performance, and the structure of the learned roles. The main remaining issues are about positioning and scope rather than core soundness: the role-level inductive bias should be distinguished more clearly from prior role-discovery work, and the evidence for broader cross-task generalization remains somewhat limited beyond the current benchmark family.